# The rise and fall of the ancient northern pike master sex-determining gene

Qiaowei Pan[1,2], Romain Feron[1,2,3], Elodie Jouanno[1], Hugo Darras[2], Amaury Herpin[1], Ben Koop[4], Eric Rondeau[4], Frederick W Goetz[5], Wesley A Larson[6], Louis Bernatchez[7], Mike Tringali[8], Stephen S Curran[9], Eric Saillant[10], Gael PJ Denys[11,12], Frank A von Hippel[13], Songlin Chen[14], J Andrés López[15], Hugo Verreycken[16], Konrad Ocalewicz[17], Rene Guyomard[18], Camille Eche[19], Jerome Lluch[19], Celine Roques[19], Hongxia Hu[20], Roger Tabor[21], Patrick DeHaan[21], Krista M Nichols[22], Laurent Journot[23], Hugues Parrinello[23], Christophe Klopp[24], Elena A Interesova[25], Vladimir Trifonov[26], Manfred Schartl[27], John Postlethwait[28], Yann Guiguen[1]*

[1]INRAE, LPGP, Rennes, France; [2]Department of Ecology and Evolution, University of Lausanne, Lausanne, Switzerland; [3]Swiss Institute of Bioinformatics, Lausanne, Switzerland; [4]Department of Biology, Centre for Biomedical Research, University of Victoria, Victoria, Canada; [5]Environmental and Fisheries Sciences Division, Northwest Fisheries Science Center, National Marine Fisheries Service, NOAA, Seattle, United States; [6]Fisheries Aquatic Science and Technology Laboratory at Alaska Pacific University, Anchorage, United States; [7]Institut de Biologie Intégrative et des Systèmes (IBIS), Université Laval, Québec, Canada; [8]Fish and Wildlife Conservation Commission, Florida Marine Research Institute, St. Petersburg, United States; [9]School of Fisheries and Aquatic Sciences, Auburn University, Auburn, United States; [10]Gulf Coast Research Laboratory, School of Ocean Science and Technology, The University of Southern Mississippi, Ocean Springs, United States; [11]Laboratoire de Biologie des organismes et écosystèmes aquatiques (BOREA), MNHN, CNRS, IRD, SU, UCN, Laboratoire de Biologie des organismes et écosystèmes aquatiques (BOREA), Paris, France; [12]Unité Mixte de Service Patrimoine Naturelle – Centre d'expertise et de données (UMS 2006 AFB, CNRS, MNHN), Muséum national d'Histoire naturelle, Paris, France; [13]Department of Biological Sciences, Northern Arizona University, Flagstaff, United States; [14]Yellow Sea Fisheries Research Institute, CAFS, Laboratory for Marine Fisheries Science and Food Production Processes, Pilot National Laboratory for Marine Science and Technology (Qingdao), Qingdao, China; [15]College of Fisheries and Ocean Sciences Fisheries, Fairbanks, United States; [16]Research Institute for Nature and Forest (INBO), Brussels, Belgium; [17]Department of Marine Biology and Ecology, Institute of Oceanography, University of Gdansk, Gdansk, Poland; [18]INRAE, GABI, Jouy-en-Josas, France; [19]GeT-PlaGe, INRAE, Genotoul, Castanet-Tolosan, France; [20]Beijing Fisheries Research Institute & Beijing Key Laboratory of Fishery Biotechnology, Beijing, China; [21]U.S. Fish and Wildlife Service, Lacey, United States; [22]Conservation Biology Division, Northwest Fisheries Science Center, National Marine Fisheries Service, National Oceanic and Atmospheric Administration, Seattle, United States; [23]Institut de Génomique Fonctionnelle, IGF, CNRS, INSERM, Univ. Montpellier, Montpellier, France; [24]INRAE, Sigenae, Genotoul Bioinfo, Toulouse, France; [25]Tomsk State University, Tomsk, Russian Federation; [26]Institute of Molecular and Cellular Biology, Siberian Branch of the Russian Academy of Sciences, Novosibirsk State

*For correspondence:
yann.guiguen@inrae.fr

Competing interests: The authors declare that no competing interests exist.

University, Novosibirsk, Russian Federation; [27]University of Wuerzburg, Developmental Biochemistry, Biocenter, 97074 Würzburg, Germany; and The Xiphophorus Genetic Stock Center, Texas State University, San Marcos, United States; [28]Institute of Neuroscience, University of Oregon, Eugene, United States

**Abstract** The understanding of the evolution of variable sex determination mechanisms across taxa requires comparative studies among closely related species. Following the fate of a known master sex-determining gene, we traced the evolution of sex determination in an entire teleost order (Esociformes). We discovered that the northern pike (*Esox lucius*) master sex-determining gene originated from a 65 to 90 million-year-old gene duplication event and that it remained sex linked on undifferentiated sex chromosomes for at least 56 million years in multiple species. We identified several independent species- or population-specific sex determination transitions, including a recent loss of a Y chromosome. These findings highlight the diversity of evolutionary fates of master sex-determining genes and the importance of population demographic history in sex determination studies. We hypothesize that occasional sex reversals and genetic bottlenecks provide a non-adaptive explanation for sex determination transitions.

## Introduction

Genetic sex determination (GSD) evolved independently and repeatedly in diverse taxa, including animals, plants, and fungi (*Tree of Sex Consortium et al., 2014*; *Beukeboom and Perrin, 2014*), but the stability of such systems varies drastically among groups. In mammals and birds, conserved male or female heterogametic sex determination (SD) systems have been maintained over a long evolutionary time with conserved master sex-determining (MSD) genes (*Marshall Graves, 2008*). In contrast, teleost fishes display both genetic and environmental sex determination (ESD), and a remarkable evolutionary lability driven by rapid turnovers of sex chromosomes and MSD genes (*Kikuchi and Hamaguchi, 2013*; *Pan et al., 2017*). These characteristics make teleosts an attractive group in which to study the evolution of SD systems.

In the past two decades, the identity of a variety of MSD genes has been revealed in teleosts thanks to advances in sequencing technologies. These findings generated new hypotheses on how the birth of MSD genes through allelic diversification or duplication with or without translocation may drive sex chromosome turnover in vertebrates (*Kikuchi and Hamaguchi, 2013*; *Pan et al., 2017*). These teleost MSD genes also provided empirical support for the 'limited option' idea that states that certain genes known to be implicated in sex differentiation pathways are more likely to be recruited as new MSD genes (*Marshall Graves and Peichel, 2010*). The majority of these recently discovered MSD genes, however, were phylogenetically scattered, making it challenging to infer evolutionary patterns and conserved themes of sex chromosomes and/or MSD gene turnovers. Although comparative studies have been accomplished in medakas, poeciliids, tilapiine cichlids, salmonids, and sticklebacks (*Kikuchi and Hamaguchi, 2013*), transitions of SD systems in relation to the fate of known MSD genes within closely related species have only been explored in medakas (*Myosho et al., 2015*) and salmonids (*Guiguen et al., 2018*).

Esociformes is a small order of teleost fishes (*Figure 1*) that diverged from their sister group Salmoniformes about 110 million years ago (Mya) and diversified from a common ancestor around 90 Mya (*Campbell et al., 2013*; *Campbell and Lopéz, 2014*). With two families (Esocidae, including Esox, Novumbra, and Dallia, and Umbridae, including Umbra) and 13 well-recognized species (*Warren et al., 2020*), Esociformes is an ecologically important group of freshwater species from the northern hemisphere (*Campbell et al., 2013*; *Campbell and Lopéz, 2014*). We demonstrated that a male-specific duplication of the anti-Müllerian hormone gene (*amhby*) is the MSD gene in Esociformes and that this gene is located in a small male-specific insertion on the Y chromosome of northern pike (*Esox lucius*) (*Pan et al., 2019*).

Here, we took advantage of the small number of Esociformes species and their relatively long evolutionary history to explore the evolution of SD in relation to the fate of this *amhby* MSD gene. Generating novel draft genome assemblies and population genomic data in multiple species of

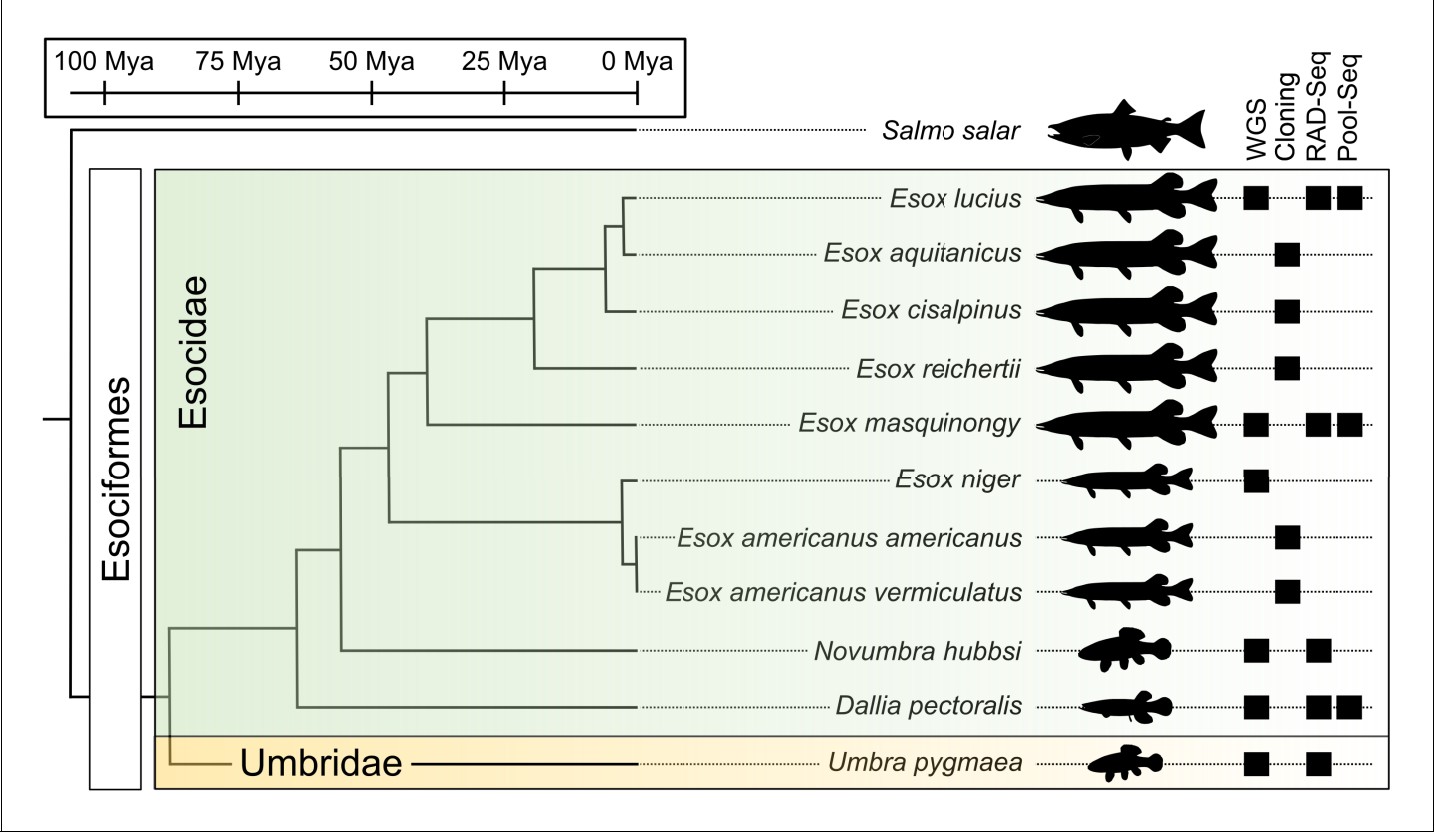

**Figure 1.** Species phylogeny with estimated divergence time (*Campbell et al., 2013*; *Kumar et al., 2017*) and technical approaches for investigating SD systems in Esociformes. The two families within Esociformes, the Esocidae and the Umbridae, are highlighted with green and yellow background, respectively. Whole-genome sequencing (WGS), homology cloning (Cloning), restriction-site-associated DNA sequencing (RAD-Seq), and pooled-sequencing (Pool-Seq) approaches used in each species are shown by a black square on the right side of the figure. Fish silhouettes were obtained from phylopic.org.

Esociformes (*Figure 1*), we traced the evolutionary trajectory of *amhby* from its origin in a gene duplication event 65–90 Mya to its demise during species- or population-specific SD transitions. Our results highlight the diverse evolutionary fates experienced by an MSD gene. Among SD transitions, we uncovered a recent loss of the entire Y chromosome in some populations of northern pike. We hypothesize that drift, exacerbated by the bottleneck effect, may have facilitated the loss of the sex chromosome along with its MSD gene in these populations, a new mechanism for entire sex chromosome loss in vertebrates.

## Results

### The *Esox lucius amhby* gene originated from an ancient duplication event

Previously, we demonstrated that a male-specific duplication of the anti-Müllerian hormone gene (*amhby*) is the MSD gene in Esociformes and that this gene is located in a small, ~140 kb male-specific insertion on the Y chromosome of northern pike (*Esox lucius*) (*Pan et al., 2019*). To explore the evolution of the autosomal copy of *amh* (*amha*) and *amhby* in Esociformes, we collected phenotypically sexed samples of most species of this order (*Figure 1* and *Table 1*). We searched for homologs of *amh* using homology cloning and whole-genome sequencing. We found two amh genes in all surveyed Esox and Novumbra species. In the more basally diverging species, *Dallia pectoralis* and *Umbra pygmaea*, we found only one *amh* gene in both species in tissue-specific transcriptome databases (*Pasquier et al., 2016*), while two *amh* transcripts were readily identified in *E. lucius* (*Pan et al., 2019*).

**Table 1.** Summary of the identification of *amhby* and its association with sex phenotypes in 11 Esociformes species, including six populations of *E. lucius* and three populations of *Esox masquinongy.*

| Species | Sampling location | Geographic region | Males with *amhby* | Females with *amhby* | p-value of *amhby* association with sex |
|---|---|---|---|---|---|
| *Esox lucius* | France | Western Europe | 161/161 (100%) | 0/60 (0%) | <2.2E-16 |
| *Esox lucius* | Sweden | Northern Europe | 20/20 (100%) | 0/20 (0%) | <2.2E-16 |
| *Esox lucius* | Poland | Central Europe | 20/20 (100%) | 0/20 (0%) | <2.2E-16 |
| *Esox lucius* | Xinjiang, China | Eastern Asia | 5/5 (100%) | 0/5 (0%) | 0.0114 |
| *Esox lucius* | Continental USA and Canada | North America | 0/88 (0%) | 0/74 (0%) | *amhby* absent |
| *Esox lucius* | Alaska, USA | North America | 17/19 (89%) | 0/19 (0%) | 1.79E-07 |
| *Esox lucius* | Siberia, Russia | Northern Asia | 9/9 (100%) | 0/8 (0%) | 0.0003 |
| *Esox aquitanicus* | France | Western Europe | 1/1 (100%) | 0/2 (0%) | 0.665 |
| *Esox cisalpinus* | Italy | Western Europe | 2/2 (100%) | 0/2 (0%) | 0.317 |
| *Esox reichertii* | China | Eastern Asia | 11/11 (100%) | 0/10 (0%) | 3.40E-05 |
| *Esox masquinongy* | Iowa, USA | North America | 18/27 (66.7%) | 4/18 (22%) | 2.01E-08 |
| *Esox masquinongy* | Quebec, Canada | North America | 9/20 (45%) | 4/20 (20%) | 0.18 |
| *Esox masquinongy* | Wisconsin, USA | North America | 57/61 (93%) | 22/61 (36%) | 1.17E-10 |
| *Esox americanus (americanus)* | Mississippi, USA | North America | 6/6 (100%) | 0/4 (0%) | 0.0123 |
| *Esox niger* | Quebec, Canada | North America | 5/5 (100%) | 0/8 (0%) | 0.0025 |
| *Novumbra hubbsi* | Washington, USA | North America | 21/23 (91%) | 0/22 (0%) | 5.28E-09 |
| *Dallia pectoralis* | Alaska, USA | North America | 0/30 (0%) | 0/30 (0%) | *amhby* absent |
| *Umbra pygmaea* | Belgium | Western Europe | 0/31 (0%) | 0/31 (0%) | *amhby* absent |

To clarify relationships among these *amh* homologs, we inferred phylogenetic trees from these sequences. These phylogenies provided a clear and consistent topology with two well-supported gene clusters among the *amh* sequences from Esocidae (*Figure 2A*, *Appendix 1—figure 1*). The first gene cluster contains the autosomal *amha* of *E. lucius* and an *amh* homolog from all other species of Esox, Dallia, and Novumbra. Therefore, we assigned these *amh* homologs as *amha* orthologs. The second gene cluster contains the Y-specific *amhby* gene of *E. lucius*, along with the other *amh* homolog from all species for which we identified two *amh* sequences. We assigned these *amh* homologs as *amhby* orthologs. The general topology of each cluster was in agreement with the Esociformes species taxonomy.

In *U. pygmaea*, the closest sister species to the Esocidae, we found a single *amh* homolog, which roots at the base of the *amha/amhby* clusters. This result indicates that the *amh* duplication happened after the divergence of Esocidae (including Esox, Dallia, and Novumbra) and Umbridae (including Umbra) lineages. In contrast to other Esocidae that have two *amh* orthologs, we found only one *amh* gene in *D. pectoralis*, which fell in the phylogeny within the *amha* ortholog cluster, suggesting that this species experienced a secondary loss of its *amhby* gene after the *amh* duplication at the root of the Esocidae lineage (*Figure 2B* and *Appendix 1—figure 1*). We confirmed the absence of an additional divergent *amh* gene in *D. pectoralis* by searching sex-specific Pool-Seq reads from 30 males and 30 females. In addition, only one *amh* homolog was found in an ongoing genome assembly project with long reads for a male *D. pectoralis* (Y. Guiguen, personal communication). Using information from an Esociformes time-calibrated phylogeny (*Campbell et al., 2013*), we estimated that the *amhby* duplicate arose between 65 and 90 Mya (*Figure 2B*).

## Sex linkage of *amhby* in the Esocidae lineage

Given that *amhby* originated from an ancient duplication event and is the MSD gene in *E. lucius*, we investigated when *amhby* became the MSD gene and whether transitions of SD systems exist in this group. Because male sex linkage of a sequence is a strong indication that this sequence is located on a Y-chromosome-specific region, we first investigated the association of *amhby* with male

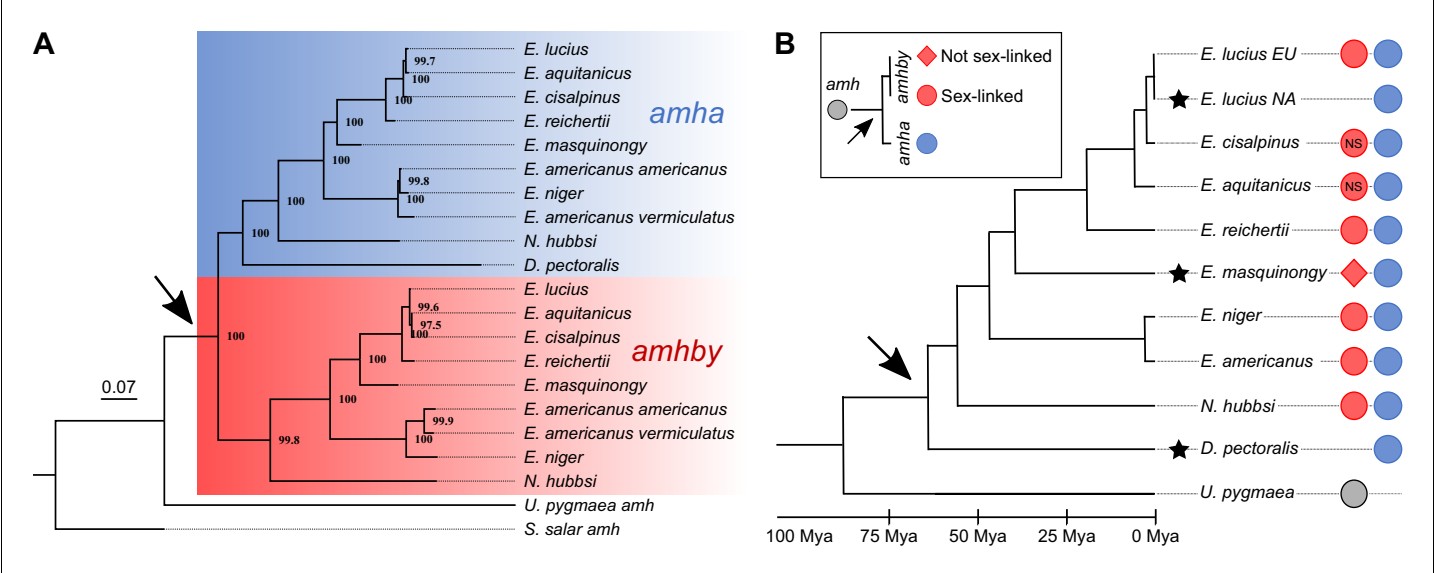

**Figure 2.** Phylogenetic analysis of *amh* homologs from the Esociformes revealed an ancient origin of *amhby*. (**A**) Phylogenetic tree of *amh* coding sequences built with the maximum-likelihood method. Bootstrap values are provided on each node of the tree. The *amha* ortholog cluster is highlighted with blue background and the *amhby* ortholog cluster is highlighted with red background. (**B**) Time-calibrated species phylogeny of the Esociformes (*Campbell et al., 2013*). The three putative SD transitions are shown by black stars. The presence of pre-duplication *amh*, as well as *amha* and *amhby* along with the *amhby* sex linkage, is represented by colored dots at the end of each branch. In *E. cisalpinus* and *E. aquitanicus*, sex linkage is not significant (NS) due to low sample size. The earliest duplication timing of *amh* is denoted by a black arrow at the root of the divergence of Esocidae.

phenotype. We found an association between male phenotype and the presence of *amhby* in most species of Esox and Novumbra (*Table 1*). This sex linkage was significant in European, Asian, and Alaskan populations of *E. lucius*, in two North American populations of *E. masquinongy* (Iowa and Wisconsin), and in all populations surveyed of *E. reichertii*, *E. americanus*, *E. niger*, and *N. hubbsi*. For two recently described species, *E. cisalpinus* and *E. aquitanicus* (*Denys et al., 2014*), we had insufficient samples with clear species and sex identification for a decisive result. We confirmed the presence of *amhby* in all males and its absence from all females in these two species, but the association was not significant due to low sample size (*Table 1*). Because *amhby* is associated with male phenotype in most species of Esox and Novumbra, it likely gained an MSD role shortly after its origin either at the root of Esocidae or before the split of Esox and Novumbra lineages (~56 Mya) (*Campbell et al., 2013*). Despite this global conservation of the linkage between *amhby* and male phenotype, however, we found some variations in *amhby* sex association across populations of *E. lucius* and *E. masquinongy* (*Table 1*). These population variations are further investigated below.

## Whole-genome analyses of the evolution of sex determination systems in Esociformes

Because *amhby* was not completely associated with male phenotype in *E. masquinongy* and *N. hubbsi* (*Table 1*) and the gene was not found in *D. pectoralis* and *U. pygmaea*, we also used population genomic approaches to search for whole-genome sex-specific signatures in these species. Because of the close phylogenetic distance (45 Mya) between *E. lucius* and *E. masquinongy*, we used the *E. lucius* genome assembly to remap reads from *E. masquinongy*. We performed Pool-Seq analysis of *E. masquinongy* (Iowa, USA) to compare the size and location of its sex locus with that of *E. lucius* (*Pan et al., 2019*). Comparison of the sex-specific heterozygosity across the entire genome revealed that a single region of less than 50 kb, containing the highest density of male-specific single-nucleotide polymorphisms (SNPs) in *E. masquinongy*, is located in a region homologous to the proximal end of LG24, where the sex locus of *E. lucius* is located. Besides LG24, no other linkage group showed enrichment for sex-biased chromosome differentiation (*Appendix 1—figure 2*),

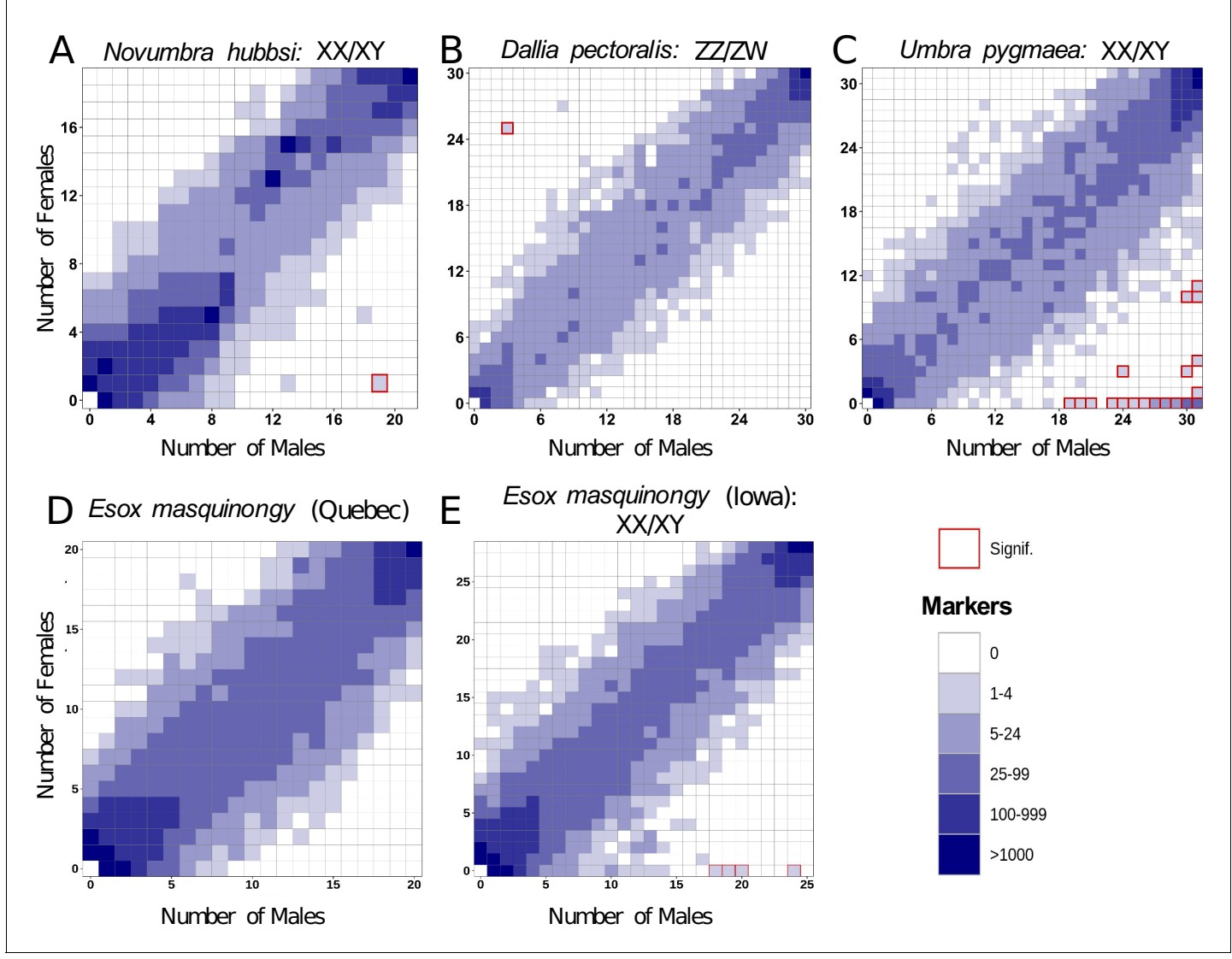

**Figure 3.** Characterization of sex determination systems in different Esociformes species through RAD-Seq analyses. Each tile plot shows the distribution of non-polymorphic RAD-Seq markers shared between phenotypic males (horizontal axis) and phenotypic females (vertical axis). The intensity of color for a tile reflects the number of markers present in the respective number of males and females. Tiles that are significantly associated with phenotypic sex (chi-square test, p<0.05 after Bonferroni correction) are highlighted with a red border. (A) In *N. hubbsi*, two markers were present in 19 of 21 males and in 1 of 19 females, indicating a significant male association and thus an XX/XY SD system with a small sex locus. (B) In *D. pectoralis*, one marker showed a significant association with females, while no marker showed association with males, indicating a ZZ/ZW SD system. (C) In *U. pygmaea*, 140 markers showed a significant association with males, while no marker showed association with females, indicating a XX/XY SD system with a large non-recombining sex locus. (D) In the population of *E. masquinongy* from Quebec (Canada), no marker was associated with either sex. (E) In a population of *E. masquinongy* from Iowa (USA), five markers were significantly associated with male phenotype, indicating a XX/XY SD system.

The online version of this article includes the following source data and source code for figure 3:

**Source code 1.** R Script to generate *Figure 3*.
**Source data 1.** Distribution of RADsex markers of *E. masquinongy* from a Quebec population with a minimal marker depth of 10 reads.
**Source data 2.** Distribution of RADsex markers of *E. masquinongy* from a Iowa population with a minimal marker depth of 10 reads.
**Source data 3.** Distribution of RADsex markers of *N. hubbsi* a minimal marker depth of 10 reads.
**Source data 4.** Distribution of RADsex markers of *U. pygmaea* a minimal marker depth of 10 reads.
**Source data 5.** Distribution of RADsex markers of *D. pectoralis* a minimal marker depth of 10 reads.

suggesting that the location and the small size of the sex locus are likely conserved between *E. lucius* and *E. masquinongy*.

No high-quality genome assembly was available from a closely related species to *N. hubbsi*. Therefore, we performed de novo RAD-Seq analysis on 21 males and 19 females for *N. hubbsi*. We found two markers showing a significant association with male sex and no female-biased marker (*Figure 3A*). This result supports the male heterogametic SD system (XX/XY) that was suggested by the *amhby* sex linkage and also indicates that the *N. hubbsi* sex locus is likely small (41.6 restriction enzyme cutting sites/Mb across the genome, on average, *Supplementary file 2*).

In *D. pectoralis* and *U. pygmaea*, two species in which *amhby* is absent, we carried out RAD-Seq analyses comparing phenotypic males and females. In *D. pectoralis*, we found only three female-biased RAD markers, suggesting that this species also has a small sex locus region (30.8 restriction enzyme cutting sites/Mb on average, *Supplementary file 2*) under a female heterogametic SD system (ZZ/ZW) (*Figure 3B*). This ZZ/ZW SD system was further supported by an independent Pool-Seq analysis revealing 45 times more female-specific k-mers (N = 1,081,792) than male-specific k-mers (N = 23,816). This excess of female-specific k-mers indicates that females carry genomic regions that are absent from males and thus that females are the heterogametic sex. In contrast, 140 male-biased and no female-biased RAD markers were identified in *U. pygmaea*, supporting that this species has a large sex locus (26.2 restriction enzyme cutting sites/Mb on average, *Supplementary file 2*) under a male heterogametic SD system (XX/XY) (*Figure 3C*).

## Some populations of *Esox lucius* lost their Y chromosome and ancestral master sex-determining gene

Although *amhby* is the MSD gene in European populations of northern pike (*Pan et al., 2019*), this gene was absent from the genome assembly (GCA_000721915.3) of a male specimen from a Canadian population (GCA_000721915.3). To explore this discrepancy, we surveyed the sex linkage of *amhby* in geographically isolated populations of *E. lucius*. We found significant male linkage in all investigated European and Asian populations and in one Alaskan population (*Table 1*). In contrast, *amhby* was absent from both males and females in all other North American populations.

To investigate whether the loss of *amhby* in most North American populations coincides with large genomic changes, we compared genome-wide sex divergence patterns between a European population carrying the *amhby* gene (Ille-et-Vilaine, France) and a North American population that lacks *amhby* (Quebec, Canada) using a Pool-Seq approach. We aligned Pool-Seq reads to an improved European *E. lucius* genome assembly (NCBI accession number SDAW00000000; assembly metrics presented in *Supplementary file 3*) in which all previously identified Y-specific contigs (*Pan et al., 2019*) were scaffolded into a single contiguous locus on the Y chromosome (LG24). In the European population, Pool-Seq analysis confirmed, with better resolution, previous results (*Pan et al., 2019*), showing the presence of a small Y chromosome region (~140 kb) characterized by many male-specific sequences at the proximal end of LG24 (*Figure 4* A.1 and 4B.1). In comparison, virtually no reads from either male or female pools from the Quebec population mapped to this 140 kb Y-specific region (*Figure 4* A.2 and 4B.2). Furthermore, we observed no differentiation between males and females along the remainder of the genome with either Pool-Seq or reference-free RAD-Seq (*Appendix 1—figure 3*). Together, these results suggest that the Quebec population and likely other mainland US populations, where the MSD gene *amhby* was not found, lack not only *amhby*, but also the surrounding Y-specific region identified in European populations. Moreover, the new sex locus of these North American populations, if it exists, is too small to be detected by the RAD-Seq and Pool-Seq approaches.

## Evolving sex determination systems: the case of the muskellunge, *Esox masquinongy*

As in *E. lucius*, we found variation in *amhby* sex linkage among different populations of *E. masquinongy* (*Table 1*). In addition, we found that two males and one female of *E. masquinongy* were heterozygous for *amhby* in the population from Quebec (*Appendix 1—figure 4*), a finding that conflicts with the expected hemizygous status of a Y-specific MSD gene. We thus used RAD-Seq to compare genome-wide patterns of sex differentiation in a population from Iowa (USA) where *amhby* was significantly associated with male phenotype and in a population from Quebec (Canada) where it was

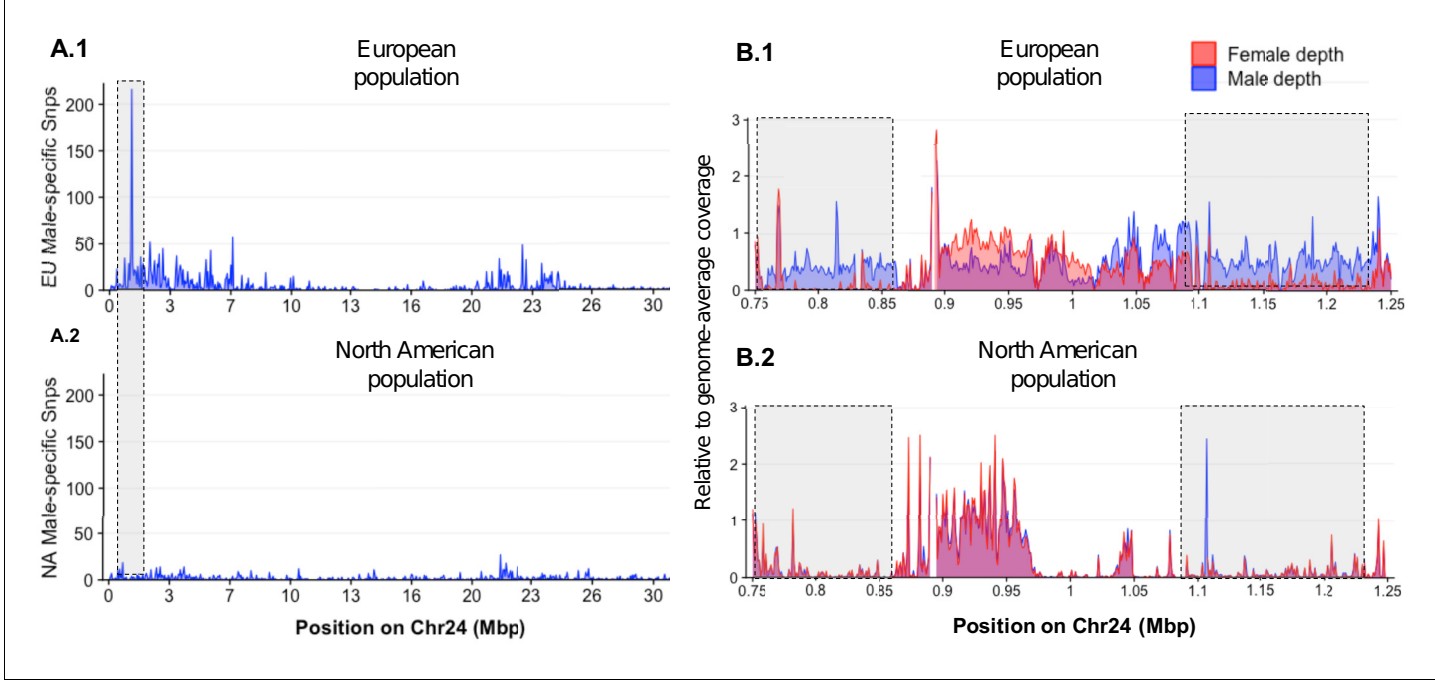

**Figure 4.** Loss of the ancestral sex locus and master sex determination gene in North American northern pike. (A) The numbers of male-specific SNPs in 50 kb non-overlapping windows deduced from Pool-Seq data of a European population (A.1) and North American population (A.2) of *E. lucius* were mapped to the Y chromosome (LG24) of the sequence of *E. lucius* from a European male genome (SDAW00000000). The greatest number of male-specific SNPs is found in a single window located at the proximal end of LG24 (0.95 Mb–1.0 Mb) that contains 182 male-specific SNPs (highlighted by the gray box) in the European *E. lucius* population. The same region, however, shows no differentiation between males and females in the North American population. (B) Relative coverage of male and female Pool-Seq reads is indicated by blue and red lines, respectively, in contigs containing *amhby* in the European population (B.1) and in the North American population (B.2). Only zoomed view on the European population sex-specific region is presented to facilitate the visualization of the differences in coverage between two populations. We searched for Y-specific regions in 1 kb windows. A window is considered Y specific if it is covered by few mapped female reads (<3 reads per kb) and by male reads at a depth close to half of the genome average (relative depth between 0.4 and 0.6). Based on these depth filters, in the European population we identified 70 potential Y-specific 1 kb windows located between 0.76 Mb and 0.86 Mb and 70 Y-specific 1 kb windows located between 1.0 Mb and 1.24 Mb on LG24. This region is not covered by male or female reads in the North American population, indicating the loss of the entire Y-specific region, highlighted by the gray boxes. The region between 0.86 Mb and 1.0 Mb is not sex specific in both population and is likely an assembly artifact.

The online version of this article includes the following source data and source code for figure 4:

**Source code 1.** R script to generate *Figure 4*.
**Source data 1.** Pool-Seq comparison of sex-specific SNPs in windows of 50 kb between males and female from a European population of *E. lucius*.
**Source data 2.** Pool-Seq comparison of sex-specific coverage in windows of 1 kb between males and female from a European population of *E. lucius*.
**Source data 3.** Pool-Seq comparison of sex-specific SNPs in windows of 50 kb between males and female from a North American population of *E. lucius*.
**Source data 4.** Pool-Seq comparison of sex-specific coverage in windows of 1 kb between males and female from a North American population of *E. lucius*.
**Source data 5.** *E. lucius* chromosome length file for the R script.

not. In the Iowa population, five RAD markers were significantly associated with male phenotype, while no marker was associated with female phenotype (*Figure 3D*). This result indicates a male heterogametic SD system (XX/XY) with a low differentiation between the X and Y chromosomes, as observed in northern pike (*Pan et al., 2019*) and *N. hubbsi* (this study). In contrast, we did not find a sex-specific marker in the Quebec population (*Figure 3E*). This result was further supported by the analysis of a publicly available dataset from another Quebec population (PRJNA512459, *Appendix 1—figure 5*), suggesting that the sex locus of *E. masquinongy* from these Quebec populations is too small to be detected with RAD-Seq (38.4 restriction enzyme cutting sites/Mb on average, *Supplementary file 2*) or that this population displays a multifactorial GSD or ESD system.

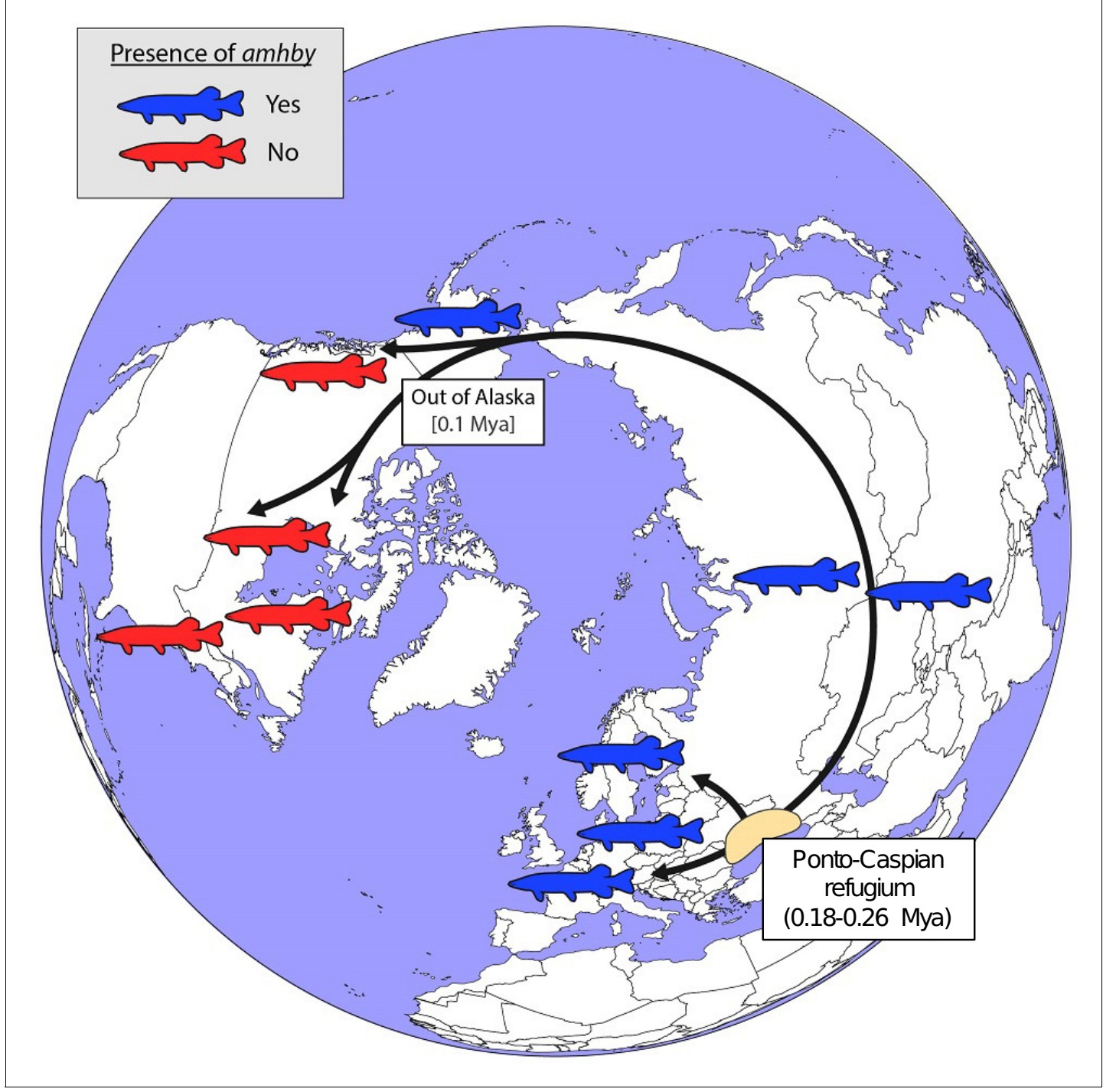

**Figure 5.** Schematic representation of a hypothesized route of post-glacial *E. lucius* expansion from a Eurasian refugium ~0.18 to 0.26 Mya. The presence (blue silhouettes)/absence (red silhouettes) of *amhby* in different populations, showing that this MSD gene was lost in some North American populations during an out-of-Alaska dispersal around 0.1 Mya. The hypothesized refugium in the Ponto Caspian region is highlighted in beige. An alternative route of post-glacial dispersal is shown in *Figure 5—figure supplement 1*.

The online version of this article includes the following figure supplement(s) for figure 5:

**Figure supplement 1.** Schematic representation of an alternative hypothesized route of post-glacial *E. lucius* expansion from a Eurasian refugium ~0.18 to 0.26 Mya.

## Evolution of the structure of the *amhby* gene in the Esocidae

The typical *amh* gene in teleost fishes comprises seven exons encoding a protein that contains 500–571 amino acids. The conserved C-terminal TGF-β domain is crucial for canonical Amh function (*di Clemente et al., 2010*). The predicted structures of most of the *amha* and *amhby* genes in Esociformes are consistent with this canonical structure, and most *amha* and *amhby* genes do not show signature of relaxation from purifying selection (*Supplementary file 4* and *Supplementary file 5*). However, in both *E. niger* and *N. hubbsi*, the predicted Amhby protein is truncated in its N- or C-terminal part. In *E. niger,* this truncated *amhby* gene is flanked by repeated elements and encompasses only three of the seven conserved Amh exons, that is exons 5, 6, and 7 with a truncated TGF-β domain (*Appendix 1—figure 6*). In *N. hubbsi*, the truncated *amhby* gene contains the seven conserved Amh exons but with an N-terminal truncation of exon 1 encoding only eight amino acids with no homology to the conserved 50 amino acid sequence of the first Amh exon of other Esocidae (*Appendix 1—figure 6*). Together, these results show that in some species where *amhby* is sex linked, the protein sequence is strongly modified in the N- and C-terminal-conserved domains that are needed for proper protein secretion and correct conformation (*di Clemente et al., 2010*), leaving questions as to whether these proteins are functional.

## Discussion

The northern pike MSD gene *amhby* substantially diverges from its autosomal copy *amha*, suggesting an ancient origin (*Pan et al., 2019*), unlike other cases of *amh* duplication in fishes, that appears to be comparatively young (*Hattori et al., 2013*; *Li et al., 2015*). Here, we show that the *amhby* duplicate emerged before the split of Esocidae and Umbridae between 65 and 90 Mya (*Campbell et al., 2013*) and was subsequently secondarily lost in the Dallia lineage. We also demonstrate that *amhby* presence is significantly associated with male phenotype in most species of Esox and in Novumbra. Although we cannot rule out the possibility that *amhby* was recruited independently as the MSD gene in Esox and Novumbra, our results suggest the more parsimonious hypothesis that *amhby* likely acquired an MSD function before the diversification of Esocidae at least 56 Mya.

Studies on SD systems of cichlid fishes and true frogs suggested that frequent turnovers can keep sex chromosomes undifferentiated (*Gammerdinger et al., 2018*; *Jeffries et al., 2018*). It was proposed that in these organisms, the turnover of sex chromosomes was facilitated by autosomes that host genes involved in sex differentiation pathways that could be readily recruited as new MSD genes, facilitating turnovers of sex chromosomes (*Vicoso, 2019*). Alternative evolutionary pathways could also involve conserved MSD genes that translocate onto different autosomes, as demonstrated for the salmonid *sdY* gene (*Guiguen et al., 2018*). Interestingly, this scenario does not seem to be the case with Esocidae. No highly differentiated sex chromosomes were observed in the surveyed population of *E. masquinongy* and *N. hubbsi* (this study), or in *E. lucius*, despite Esocidae having retained the same ancestral XX/XY system (likely controlled by *amhby*) for at least 56 million years. In addition, whole-genome analyses in *E. masquinongy* suggest that its male-specific locus on the Y chromosome is conserved in terms of size and location with that of *E. lucius* despite 45 million years of divergence. Such conservation is unusual in teleosts, where frequent turnover of SD systems is considered the norm (*Kikuchi and Hamaguchi, 2013*; *Pan et al., 2017*). In line with our present results, substantially undifferentiated sex chromosomes have also been maintained over relatively long evolutionary periods in some Takifugu fish species (*Kamiya et al., 2012*), supporting the idea that theoretical models of sex chromosome evolution cannot be generalized across taxa and that additional, as-yet unknown evolutionary forces can prevent sex chromosome decay.

Even though this MSD gene has been maintained for a long evolutionary period, our results revealed that it has been subjected to several independent SD transitions, potentially driven by different mechanisms in the Esocidae lineage. In *D. pectoralis*, we observed a transition from the ancestral XX/XY male heterogametic system shared by Esocidae and *U. pygmaea* to a ZZ/ZW female heterogametic system. This transition also coincided with the loss of *amhby*. This shift from male to female heterogamety could have been driven by the rise of a new dominant female determinant, leading to the obsolescence and the subsequent loss of the ancestral male MSD gene, similar to the transition of SD documented among the Oryzias and also in the housefly (*Kozielska et al., 2008*, *Myosho et al., 2012*).

The other SD transitions that we characterized, those in *E. lucius* and *E. masquinongy*, are probably more recent because these transitions are found within populations of the same species. In *E. masquinongy*, we observed population-specific variation in SD systems, supporting the conservation of an XX/XY system with *amhby* as the MSD gene in two US populations belonging to the northern lineage, but not in a Quebec population belonging to the Great Lakes lineage (*Turnquist et al., 2017*). This result suggests that although *amhby* is still present in the genome of the Great Lakes lineage, it is no longer the MSD gene and that a new population-specific SD mechanism emerged after the split of the lineages. Previously, female heterogamety was inferred for the Great Lakes lineage based on the presence of males in the offspring of gynogenetic females (*Dabrowski et al., 2000*). With RAD-Seq, we did not find evidence for female-specific genomic regions in the Great Lakes lineage of *E. masquinongy*. Higher resolution approaches are needed to identify differentiation between male and female genomes to reveal the new SD locus and its interaction with *amhby* in the Great Lakes lineage.

In *E. lucius*, we found that *amhby* is present and completely sex linked in European, Asian, and Alaskan regions, but is not present elsewhere in North America (*Figure 5*). This finding suggests that *amhby* functions as an MSD gene in Eurasian and Alaskan populations, but that populations from elsewhere in North America lost *amhby* together with the entire sex locus. The current geographic distribution of *E. lucius* results from a post-glacial expansion from three glacial refugia ~0.18 to 0.26 Mya. The North American populations lacking *amhby* belong to a monophyletic lineage with a circumpolar distribution originating from the same refugia as the other population we surveyed that carry *amhby* (*Skog et al., 2014*). Fossil records from Alaska support the idea of an 'out of Alaska' North American expansion of *E. lucius* within the last 100,000 years (*Wooller et al., 2015*), suggesting that the Y-specific sequences containing *amhby* were lost during this dispersal period (*Figure 5*, *Appendix 1—figure 7*).

The loss of the entire sex locus in such a short evolutionary time is unlikely to have resulted from the pseudogenization of the *amhby* gene, as is the case for the ancestral MSD gene of the Luzon medaka (*Myosho et al., 2012*). Rather, it is probably the result of colonization by a small pool of females and sex-reversed XX males. Such a founder-effect hypothesis is supported by the fact that North American populations of *E. lucius* display a much lower genetic diversity compared to other populations from the same lineage (*Skog et al., 2014*). This hypothesis is also supported by the fact that environmental influence on GSD systems is a well-documented phenomenon in fishes (*Baroiller et al., 2009*; *Chen et al., 2014*; *Sato et al., 2005*) and that XX sex reversal induced by environmental factors has been observed occasionally in captive *E. lucius* (*Pan et al., 2019*). Our whole-genome analyses failed to identify any sex-associated signals in these North American populations, meaning that if such a new sex locus exists, it would lack detectable signatures of sequence differentiation, similar to the one-SNP sex locus of some Takifugu species (*Kamiya et al., 2012*). Whether ESD, which is common among poikilothermic animals including teleosts (*Navara, 2018*), could be the SD system in these North American populations or whether these populations already have a new GSD system in place is still unresolved. Notably, this Y chromosome loss in natural populations of *E. lucius* mirrors the loss of the W chromosome in lab strains of zebrafish after just a few decades of selective breeding (*Wilson et al., 2014*).

A few evolutionary models have been formulated to capture the dynamics of sex chromosome turnover (*Vicoso, 2019*). The replacement of the ancestral SD locus by a new one could be driven by drift alone, by positive selection (*Bull and Charnov, 1977*), by sexual antagonistic selection (*van Doorn and Kirkpatrick, 2007*), or by the accumulation of deleterious mutations at the sex locus (*Blaser et al., 2013*). In all of these models, the ancestral sex chromosomes would revert to autosomes only when the new SD locus is fixed in the population. However, in North American populations of *E. lucius*, the ancestral sex chromosome was likely lost due to drift in bottlenecked populations. Interestingly, this process would not require the simultaneous emergence of a new GSD system, given the flexibility of SD mechanisms in teleosts, where environmental cues generate phenotypic sexes in the absence of a GSD system. ESD, which is easily invaded by a new genetic system (*Muralidhar and Veller, 2018*; *van Doorn, 2014*), may serve as a transitional state between sex chromosome turnovers.

We found two cases of altered *amhby* copies. In *N. hubbsi* and *E. niger*, *amhby* is strongly sex linked, which suggests that it is located in a Y-specific region of the genome. In both species, however, the *amhby* gene lacks parts of the conserved C- and/or N-terminal Amh regions, which are

needed for proper protein secretion and interaction between Amh and its receptor (*di Clemente et al., 2010*). Although we cannot exclude that these are non-functional *amhby* copies, this hypothesis seems unlikely because the tight sex linkage of *amhby* in these species would require the independent emergence of a new MSD gene in close proximity to *amhby*. It is more likely that these copies still function as MSD genes, but with altered protein sequences. A similar example of a truncated *amh* duplicate acting as an MSD gene is found in the teleost cobalt silverside *Hypoatherina tsurugae* (*Bej et al., 2017*). Additional cases of truncated duplicated genes functioning as MSD genes have been described in Salmonids, yellow perch (*Perca flavescens*), and Atlantic herring (*Clupea harengus*), suggesting that the preservation of ancestral domains is not necessary for a duplicated protein to assume a novel role (*Feron et al., 2020b*; *Rafati et al., 2020*; *Yano et al., 2012*). These cases are all consistent with domain gains and losses contributing to new protein functions (*Moore and Bornberg-Bauer, 2012*).

Collectively, our results depict the evolutionary trajectories of a conserved MSD gene in an entire order of vertebrates, highlighting both the potential stability of MSD genes as well as non-differentiated sex chromosomes in some lineages, and the dynamics of species- or population-specific SD evolution in teleost fishes. Our results highlight the importance of careful consideration of the population demographic history of SD systems, and of the potential buffering role of ESD during transitions between genetic SD systems in sex chromosome evolution.

## Materials and methods

### Sample collection
Information on the Esociformes species, collectors, sexing method, and experiments is provided in *Supplementary file 6*.

### Genomic DNA extraction
Fin clips were preserved in 75–100% ethanol at 4°C until genomic DNA (gDNA) extraction. For genotyping, samples were lysed with 5% Chelex and 25 mg of Proteinase K at 55°C for 2 hr, followed by incubation at 99°C for 10 min. For Illumina sequencing, gDNA was obtained using NucleoSpin Kits for Tissue (Macherey-Nagel, Duren, Germany) following the producer's protocol. DNA concentration was quantified using Qubit dsDNA HS Assay Kit (Invitrogen, Carlsbad, CA) and a Qubit3 fluorometer (Invitrogen, Carlsbad, CA). For Pool-Seq, DNA from different samples was normalized to the same quantity before pooling for male and female libraries separately. High-molecular-weight gDNA for long-read sequencing was extracted as described by *Pan et al., 2019*.

### Genome and population genomics sequencing
Draft genomes of northern pike, muskellunge (*E. masquinongy*), chain pickerel (*E. niger*), Olympic mudminnow (*N. hubbsi*), and Alaska blackfish (*D. pectoralis*) were sequenced using a whole-genome shotgun strategy with 2 × 250 bp Illumina reads. Libraries were built using the Truseq nano kit (Illumina, FC-121–4001) following instructions from the manufacturer. gDNA (200 ng) was briefly sonicated using a Bioruptor (Diagenode). The gDNA was end-repaired and size-selected on beads to retain fragments of ~550 bp, and these fragments were a-tailed and ligated to Illumina's adapter. The ligated gDNA was then subjected to eight PCR cycles. Libraries were checked on a fragment analyzer (AATI) and quantified by qPCR using a library quantification kit from KAPA. Libraries were sequenced on a HIseq2500 using the paired end 2 × 250 bp v2 rapid mode according to the manufacturer's instructions. Image analysis was performed by HiSeq Control Software, and base calling was achieved using Illumina RTA software. The output of each run is presented in *Supplementary file 7*. To improve the European male genome of *E. lucius*, we generated an extra coverage of Oxford Nanopore long reads using a higher fragment size (50 kb) library made from gDNA extracted from a different male from the same European population as the sample used for the previous genome assembly. Library construction and genome sequencing were carried out as in *Pan et al., 2019* and 12.7 Gbp of new data were generated from one PromethION flowcell.

Pool-Seq was performed on *E. masquinongy*, *D. pectoralis*, and the North American populations of *E. lucius*. Pooled libraries were constructed using a Truseq nano kit (Illumina, FC-121–4001) following the manufacturer's instructions. Male and female DNA Pool-Seq libraries were prepared for each

species using the Illumina TruSeq Nano DNA HT Library Prep Kit (Illumina, San Diego, CA) with the same protocol as for the draft genome sequencing. The libraries were then sequenced on a Nova-Seq S4 lane (Illumina, San Diego, CA) using the paired-end 2 × 150 bp mode with Illumina NovaSeq Reagent Kits following the manufacturer's instructions. The output of each run for each sex is presented in *Supplementary file 7*.

RAD-Seq was performed on *E. masquinongy*, *N. hubbsi*, *D. pectoralis*, *U. pygmaea*, and the North American populations of *E. lucius*. RAD libraries were constructed from gDNA extracted from fin clips for each species using a single *Sbf1* restriction enzyme as in *Amores et al., 2011*. Each library was sequenced on one lane of an Illumina HiSeq 2500. The summary of the output of each dataset is presented in *Supplementary file 8*.

## Analysis of population genomic data for sex-specific signals

Raw RAD-seq reads were demultiplexed with the *process_radtags.pl* script from *stacks* version 1.44 (*Catchen et al., 2013*) with all parameters set to default. Demultiplexed reads for each species were analyzed with the *RADSex* computational workflow (*Feron et al., 2020a*) using the *radsex* software version 1.1.2 (10.5281/zenodo.3775206). After generating a table of markers depth with *process*, the distribution of markers between sexes was computed with *distrib* and markers significantly associated with phenotypic sex were extracted with *signif* using a minimum depth of 10 (−d 10) for both commands and with all other settings at the default. RAD-Seq figures were generated with *sgtr* (10.5281/zenodo.3773063).

For each Pool-Seq dataset, reads were aligned to the reference genome using *BWA mem* (version 0.7.17) (*Li and Durbin, 2009*). Alignment results were sorted by genomic coordinates using *samtools sort* (version 1.10) (*Li et al., 2009*), and PCR duplicates were removed using *samtools rmdup*. A file containing nucleotide counts for each genomic position was generated with the *pileup* command from PSASS (version 3.0.1b, 10.5281/zenodo.3702337). This file was used as input to compute $F_{ST}$ between males and females, number of male- and female-specific SNPs, and male and female depth in a sliding window along the entire genome using the *analyze* command from PSASS with parameters: window-size 50,000, output-resolution 1000, freq-het 0.5, range-het 0.15, freq-hom 1, range-hom 0.05, min-depth 1, and group-snps. The analysis was performed with a snakemake workflow available at https://github.com/SexGenomicsToolkit/PSASS-workflow. Pool-Seq figures were generated with *sgtr* version 1.1.1 (10.5281/zenodo.3773063).

For k-mer analysis, 31-mers were identified and counted in the reads of the male and female pools using the '*count*' command from *Jellyfish* (version 2.2.10) (*Marçais and Kingsford, 2011*) with the option '-C' activated to count only canonical k-mers and retaining only k-mers with an occurrence >5 and <50,000,000. Tables of k-mer counts produced by *Jellyfish* were merged with the 'merge' command from *Kpool* (https://github.com/INRA-LPGP/kpool), and the resulting merged table was filtered using the 'filter' command to only retain k-mers present >25 times in one sex and <5 times in the other sex; such k-mers were considered sex biased.

## Sequencing of *amha* and *amhby* genes

To survey the presence of *amha* (the canonical copy of *amh*) and *amhby* in Esociformes, we collected samples of 11 species (*Table 1*): all species in the genus Esox (*E. americanus americanus*, *E. americanus vermiculatus*, *E. aquitanicus*, *E. cisalpinus*, *E. masquinongy*, *E. reichertii*, and *E. niger*), *N. hubbsi* (the only species in its genus), *D. pectoralis* (the only well-recognized species in its genus), and one species from the genus Umbra (*U. pygmaea*). To search for *amh* homologs in the genomes of these 11 species, we either sequenced PCR amplicons with custom primers (*Supplementary file 9*) and/or sequenced and assembled the genome (*Supplementary file 10*) and searched in the assembly and in the raw reads for the presence of one or two *amh* genes.

For species closely related to *E. lucius* (*E. aquitanicus*, *E. cisalpinus*, and *E. reichertii*), *amh* homologs were amplified with primers designed on *amha* or *amhby* of *E. lucius*. Complete sequences of *amh* homologs were obtained from overlapping amplicons covering the entire genomic regions of both *amha* and *amhby* with primers anchored from upstream of the start codon (SeqAMH2Fw1 and SeqAMH1Fw1) and downstream of the stop codon (SeqAMH2Rev3 and SeqAMH1Rev4). All PCR amplicons were then Sanger sequenced from both directions and assembled to make consensus gene sequences.

For three other Esocidae species (*E. niger*, *E. masquinongy*, *N. hubbsi*), only partial amplifications of *amh* sequences were obtained with primers designed on *amha* or *amhby* of *E. lucius*. To acquire the complete *amh* sequences, we generated draft genome assemblies from *amhby*-carrying individuals.

For the two species from the more divergent genera (*D. pectoralis* and *U. pygmaea*), we were unable to amplify *amhby* with primers designed on regions that appear conserved on *Esox* species. Therefore, we also generated a genome assembly from a phenotypic male of each species.

For *E. niger*, *E. masquinongy*, *N. hubbsi*, and *D. pectoralis*, we used *amha* and *amhby* genomic sequences from *E. lucius* as queries and searched for *amh* homologs with *blastn* (https://blast.ncbi.nlm.nih.gov/Blast.cgi, version 2.10.0+) (*Altschul et al., 1997*) using the parameters 'Max target sequences'=50 and 'Max matches in a query range'=1.

For *U. pygmaea*, the most divergent species from *E. lucius* in this study, blasting with *E. lucius amha* and *amhby* sequences did not yield any result. We used the *blastn* strategy with the coding sequence of *amh* as well as tblastn of the protein sequence from *Salmon salar*, which returned only one contig.

No genome was available for the other two more distantly related species of *E. lucius*, *E. americanus americanus*, and *E. americanus vermiculatus*; therefore, we designed primers on regions conserved in multi-sequence alignments of *amha* and *amhby* in the other Esox species (*Supplementary file 9*).

To investigate whether truncations of Amhby in *E. niger* and *N. hubbsi* could be assembly artifacts, we searched for potential missing homologous sequences in raw genome reads of both species using *tblastn* (https://blast.ncbi.nlm.nih.gov/Blast.cgi, version 2.10.0+) (*Altschul et al., 1997*) using the exon 1 protein sequence from *E. lucius* as a query. Homologous sequences were determined with a maximum e-value threshold of 1E-10.

## Validating *amh* gene number in *D. pectoralis* and *U. pygmaea*

To check whether the two copies of *amh* were not collapsed into a single sequence during assembly, we computed the total number of apparent heterozygous sites in the *amh* region in population genomic data with the expectation that the presence of two divergent gene copies should result in high apparent heterozygosity when remapped on the single copy assembled in the genome. We compared these heterozygosity data in species where we only identified one *amh* gene, that is *D. pectoralis* and *U. pygmaea*, to results observed when mapping sex-specific pooled libraries of *E. lucius* (n = 30 males and 30 females, respectively) to a female reference genome containing only *amha* (GCA_004634155.1). Because there are two *amh* genes in male genomes of *E. lucius* and one *amh* gene in female genomes, this result from mapping *E. lucius* sex-specific Pool-Seq reads serves as a reference for expected number of variant sites. Reads from the male and female pools of *E. lucius* and from male pool of *D. pectoralis* were aligned separately to the reference genome (GCA_004634155.1) and our draft genome, respectively, using *BWA mem* version 0.7.17 (*Li and Durbin, 2009*) with default parameters. Each resulting BAM file was sorted and PCR duplicates were removed using *Picard* tools version 2.18.2 (http://broadinstitute.github.io/picard) with default parameters. Variants were then called for each BAM file using *bcftools mpileup* version 1.9 (*Li et al., 2009*) with default parameters on genomic regions containing *amh*, which locates between 12,906,561 and 12,909,640 bp on LG08 of *E. lucius* (GCA_004634155.1: CM015581.1), between 23,447 and 25,910 bp on the flattened_line_2941 contig of the draft genome of *D. pectoralis*, and between 760,763 and 757,962 bp on contig 633485 of our draft genome of *U. pygmaea*.

## Gene phylogenetic analysis

Phylogenetic reconstructions were performed on all *amh* homolog sequences obtained from Esociformes with the *S. salar amh* used as an outgroup. Full-length CDS were predicted with the *FGENESH+* (*Solovyev et al., 2006*) suite based on the genomic sequence and Amh protein sequence from *E. lucius*. Sequence alignments were performed with *MAFFT* (version 7.450) (*Katoh and Standley, 2013*). Both maximum-likelihood and Bayesian methods were used for tree construction with *IQ-TREE* (version 1.6.7) (*Minh et al., 2020*) and *Phylobayes* (version 4.1) (*Lartillot et al., 2007*), respectively.

Additional analyses were carried with or without the truncated *amh*/Amh sequences of *E. niger* and *N. hubbsi* using the *amh*/Amh homolog of *S. salar* as an outgroup. CDS and proteins were predicted with the *FGENESH+* suite (*Solovyev et al., 2006*), based on genomic sequences. These putative CDS and protein sequences were then aligned using *MAFFT* (version 7.450) (*Katoh and Standley, 2013*). Residue-wise confidence scores were computed with *GUIDANCE 2* (*Sela et al., 2015*), and only well-aligned residues with confidence scores above 0.99 were retained. The resulting alignment file was used for model selection and tree inference with *IQ-TREE* (version 1.6.7) (*Minh et al., 2020*) with 1000 bootstraps and the 1000 SH-like approximate likelihood ratio test for robustness. To verify that the topology of the single *amh* homolog of *D. pectoralis* was not an artifact of long-branch attraction, we also constructed phylogenetic trees with only the first and second codons of the coding sequences (*Lemey et al., 2009*) as well as with complete protein sequences. For additional confirmation of the tree topology for the *amh* homologs, the same alignments were run in a Bayesian framework implemented in Phylobayes (version 4.1), (*Lartillot et al., 2007*; *Lartillot and Philippe, 2006*, *Lartillot and Philippe, 2004*) using the CAT-GTR model with default parameters, and two chains were run in parallel for approximately 1000 cycles until the average standard deviation of split frequencies remained ≤0.01.

## Detection of signatures of selection on *amha* and *amhby* genes

To compare selection pressure between *amha* and *amhby* genes, dN/dS ratios were computed between each ortholog sequence and the *amh* sequence of *U. pygmaea*, which diverged from the other Esociformes prior to the *amh* duplication. Sequences from *E. niger* and *N. hubbsi* were excluded because their *amhby* orthologs were substantially shorter and could introduce bias in the analysis. The ratio of nonsynonymous to synonymous substitution (dN/dS = $\omega$) among *amha* and *amhby* sequences was estimated using *PALM* (version 4.7) (*Yang, 2007*) based on aligned full-length CDS, and the phylogenetic tree obtained with the CDS was used in the analysis. First, a relative rate test on amino acid substitution (*Hughes, 1994*) was performed between *amha* and *amhby* pairs with the *amh* of *U. pygmaea* used as an outgroup sequence. For each species with *amh* duplication, $\omega$ was calculated between the *amha* ortholog and *amh* of *U. pygmaea* and was compared to the $\omega$ calculated between the *amhby* ortholog and *amh* of *U. pygmaea*. A Wilcoxon test was used to compare the resulting $\omega$ values for the *amh* paralogs of these species. Then, several branch and site models (M0, M1a, M2a, M7, M8, free-ratio, and branch-site) implemented in the *CODEML* package were fitted to the data. The goodness of fit of these models was compared using the likelihood ratio test implemented in *PALM*.

## Genome assembly

Raw Illumina sequencing reads for *D. pectoralis*, *E. masquinongy*, *E. niger*, and *N. hubbsi* were assembled using the *DISCOVAR* de novo software with the following assembly parameters: MAX_MEM_GB = 256, MEMORY_CHECK = False, and NUM_THREADS = 16. Assembly metrics were calculated with the assemblathon_stats.pl script (*Earl et al., 2011*), and assembly completeness was assessed with *BUSCO* (version 3.0.2) (*Simão et al., 2015*) using the Actinopterygii gene set (*Supplementary file 10*).

To facilitate the comparison of sex loci in different populations of *E. lucius*, we improved the continuity of the previously published assembly of an *E. lucius* European male (GenBank assembly accession: GCA_007844535.1). This updated assembly was performed with *Flye* (version 2.6) (*Kolmogorov et al., 2019*) using standard parameters and genome-size set to 1.1 g to match theoretical expectations (*Hardie and Hebert, 2004*). Two rounds of polishing were performed with *Racon* (version 1.4.10) (*Vaser et al., 2017*) using default settings and the Nanopore reads aligned to the assembly with *minimap2* (version 2.17) (*Li, 2018*) with the 'map-ont' preset. Then, three additional rounds of polishing were performed with *Pilon* (version 1.23) (*Walker et al., 2014*) using the 'fix all' setting and the Illumina reads aligned to the assembly with BWA mem (version 0.7.17) (*Li and Durbin, 2009*). Metrics for the resulting assembly were calculated with the assemblathon_stats.pl script (*Earl et al., 2011*). The completeness of the assembly was assessed with *BUSCO* (version 3.0.2) (*Simão et al., 2015*) using the Actinopterygii gene set (4584 genes) and the default gene model for Augustus. The resulting assembly was scaffolded with *RaGOO* (version 1.1) (*Alonge et al.,*

*2019*) using the published female genome (GenBank accession: GCA_011004845.1) anchored to chromosomes as a reference.

## Acknowledgements

We are grateful to the fish facility of INRAE LPGP for support in experimental installation and fish maintenance and to the GenoToul bioinformatics platform Toulouse Midi-Pyrenees (Bioinfo Geno-Toul) for providing help, computing, and storage resources. This work was supported by the Agence Nationale de la Recherche, the Deutsche Forschungsgemeinschaft (ANR/DFG, PhyloSex project, 2014–2016, SCHA 408/10–1, to YG and MS), and the U.S. National Institutes of Health (R01GM085318 to JHP). The MGX and Get-Plage core sequencing facilities were supported by France Genomique National infrastructure, funded as part of the 'Investissement d'avenir' program managed by the Agence Nationale pour la Recherche (contract ANR-10-INBS-09). We thank the MNHN curators for providing access to the collections of *E. aquitanicus* and *E. cisalpinus* and for the ONEMA. We thank GB Delmastro for help with specimen sampling. We also thank Mackenzie Garvey and Penny Swanson for assistance with dissection of *N. hubbsi*.

## Additional information

### Funding

| Funder | Grant reference number | Author |
|---|---|---|
| Agence Nationale de la Recherche | ANR-13-ISV7-0005 | Yann Guiguen |
| Deutsche Forschungsgemeinschaft | | Manfred Schartl |
| Agence Nationale de la Recherche | ANR-10-INBS-09 | Celine Roques Laurent Journot |
| Korea National Institute of Health | R01GM085318 | John Postlethwait |
| ANR | SCHA 408/10–1 | Yann Guiguen Manfred Schartl |

The funders had no role in study design, data collection and interpretation, or the decision to submit the work for publication.

### Author contributions

Qiaowei Pan, Conceptualization, Data curation, Formal analysis, Investigation, Visualization, Methodology, Writing - original draft; Romain Feron, Conceptualization, Data curation, Software, Formal analysis, Investigation, Visualization, Methodology, Writing - original draft; Elodie Jouanno, Software, Formal analysis, Investigation, Visualization; Hugo Darras, Investigation, Visualization, Writing - review and editing; Amaury Herpin, Supervision, Investigation, Visualization; Ben Koop, Resources, Supervision, Investigation; Eric Rondeau, Frederick W Goetz, Wesley A Larson, Louis Bernatchez, Mike Tringali, Stephen S Curran, Eric Saillant, Gael PJ Denys, Frank A von Hippel, Songlin Chen, J Andrés López, Hugo Verreycken, Konrad Ocalewicz, Celine Roques, Hongxia Hu, Roger Tabor, Patrick DeHaan, Krista M Nichols, Laurent Journot, Hugues Parrinello, Vladimir Trifonov, Resources, Investigation; Rene Guyomard, Camille Eche, Resources, Formal analysis, Investigation; Jerome Lluch, Investigation; Christophe Klopp, Elena A Interesova, Resources, Software, Investigation; Manfred Schartl, Conceptualization, Resources, Funding acquisition, Investigation, Project administration, Writing - review and editing; John Postlethwait, Conceptualization, Funding acquisition, Investigation, Project administration, Writing - review and editing; Yann Guiguen, Conceptualization, Supervision, Funding acquisition, Investigation, Visualization, Project administration, Writing - review and editing

## Author ORCIDs

Qiaowei Pan [iD] https://orcid.org/0000-0002-8501-0405
Amaury Herpin [iD] http://orcid.org/0000-0002-0630-4027
Frank A von Hippel [iD] http://orcid.org/0000-0002-9247-0231
Hugo Verreycken [iD] http://orcid.org/0000-0003-2060-7005
Christophe Klopp [iD] http://orcid.org/0000-0001-7126-5477
Elena A Interesova [iD] http://orcid.org/0000-0002-1148-6283
Yann Guiguen [iD] https://orcid.org/0000-0001-5464-6219

## Decision letter and Author response

Decision letter https://doi.org/10.7554/eLife.62858.sa1
Author response https://doi.org/10.7554/eLife.62858.sa2

# Additional files

## Supplementary files

• Source code 1. R script to generate *Appendix 1—figure 3*.

• Source data 1. Poolseq comparison of sex-specific coverage in windows of 1 kb between males and female from a North American population of *E. lucius*.

• Source data 2. Distribution of RADsex markers of a Canadian population of *E. lucius* with a minimal marker depth of 10 reads.

• Source data 3. Distribution of RADsex markers of a second Canadian population of *E. lucius* with a minimal marker depth of 10 reads.

• Source data 4. Poolseq comparison of sex-specific SNPs in windows of 50 kb between males and female from a Canadian population of *E. lucius*.

• Source data 5. Poolseq comparison of sex-specific SNPs in windows of 50 kb between males and female from an Iowa population of *E. masquinongy*.

• Source data 6. Poolseq comparison of sex-specific coverage in windows of 1 kb between males and female from an Iowa population of *E. masquinongy*.

• Source data 7. *E. masquinongy* chromosome length file for the R script.

• Supplementary file 1. Number of heterozygous sites on the amh region from Pool-seq reads (*E. lucius* and *D. pectoralis*) and whole-genome sequencing reads (*U. pygmaea*). To verify that the two copies of *amh* were not collapsed into a single sequence during assembly, we computed the total number of apparent heterozygous sites on the *amh* region in population genomic data with the expectation that the presence of two divergent gene copies should result in high apparent heterozygosity when remapped on the single copy assembled in the genome. In total, 106 variants were observed in the pool of *E. lucius* males on the *amha* region located between 12,906,561 bp and 12,909,640 bp on LG08 of *E. lucius* (GCA_004634155.1: CM015581.1), resulting from the mapping of reads originating from both *amha* and *amhby*, while only 12 variants (true allelic variations) were observed in the same region when mapping reads from the female pool originating only from *amha*. With male Pool-Seq reads from *D. pectoralis*, we observed only four variant sites on the ~3 kb *amh* region located between 23,447 bp and 25,910 bp on the flattened_line_2941 contig and zero variant sites from the female Pool-Seq reads. Compared to the 'control' of *E. lucius* Pool-Seq reads where one *amh* gene from female pool result in 12 variant sites and two *amh* gene from the male pool result in 106 variant sites, the low number of variant sites in both male and female pool of *D. pectoralis* support that only one *amh* gene is present in the genome of *D. pectoralis* regardless of the phenotypic sex of the individuals. Although sex-specific Pool-Seq reads were not available for *U. pygmaea*, we performed the same analysis with reads from the single male individual used to assemble the genome. No variant was observed in the ~3 kb region containing *amh* located between 760,763 bp and 757,962 bp on contig 633485 of our draft genome of *U. pygmaea*, supporting that only one *amh* gene is also present in this species.

• Supplementary file 2. Estimated number of *SbfI* cutting sites and RAD-Seq marker frequency estimated for *E. lucius*, *E. masquinongy*, *N. hubbsi*, *D. pectoralis*, and *U. pygmaea* based on the size of

draft genome assembly. To help estimate the size of the sex locus from sex-specific RAD markers, we determined the number of potential *SbfI* cleavage sites based on our draft genome assemblies for each species. For *E. lucius* (Canadian population), *E. masquinongy* (Iowa population), *N. hubbsi*, and *D. pectoralis*, we predicted the number of RAD-Seq cleavage sites present in each genome by counting the number of unambiguous matches for sequence of *SbfI* (CCTGCAGG), the restriction enzyme used in RAD-Seq library preparation (*Herrera et al., 2015*). On average, we found 31.8 RAD markers per Mb in *E. lucius*, 38.4 in *E. masquinongy*, 41.6 in *N. hubbsi*, 30.8 in *D. pectoralis*, and 26.2 in *U. pygmaea*. Because we do not see large species differences (26–40 RAD markers/Mb) this suggests that, apart from potential local variations of the RAD markers density in sex loci, our RAD-Seq comparative analysis could to some extent be used to compare sex locus size within species on the basis of this number of sex-specific RAD markers. We are aware of the limitation that using the number of sex-specific marker usually lead to an overestimation of the size of the sex locus. For all of our species with the exception of *U. pygmaea*, we identified very few sex-specific markers, indicating very small sex locus. This simple calculation is only intended to helps provide a rough approximation of the size of the sex locus.

- Supplementary file 3. Assemblathon and BUSCOs metrics for the new genome assembly with additional Nanopore reads of a genetic European male of *E. lucius*.

- Supplementary file 4. dN/dS ratio between the *amh* paralogs in different Esociformes and *amh* of *U. pygmaea*.

- Supplementary file 5. Log-likelihood of different selection models tested on *amha* and *amhby* orthologs of the Esociformes.

- Supplementary file 6. Information on the different Esociformes species, sample collectors, sexing method, and experiments performed in this study. *Samples from (*Ouellet-Cauchon et al., 2014*). **Sex was recorded in this *E. masquinongy* population based on the urogenital pores morphology. ***Sex was recorded in *N. hubbsi* based on the specific coloration of males during the breeding season. NR = phenotypes not recorded. NA = not applicable (sex phenotypes not recorded). Museum collection numbers are as follows: MNHN 2014–2719, MNHN 2014–2720, MNHN 2014–2721, MNHN 2014–2722, and MNHN 2014–2723 for *E. cisalpinus* and MNHN 2013–1246, MNHN 2013–1245, and MNHN 2013–838 for *E. aquitanicus*.

- Supplementary file 7. Sequencing information for the Pool-Seq and whole-genome sequencing (WGS) performed in this study.

- Supplementary file 8. Total number of reads and markers and range of markers among individuals for each RAD-Seq dataset. The number of markers retained correspond to the number of markers present with depth higher than min. depth in at least one individual.

- Supplementary file 9. Primers used in this study to amplify *amha* and *amhby* sequences from the Esociformes.

- Supplementary file 10. Assemblathon and BUSCOs metrics for draft genome assembly for the Esociformes species.

- Transparent reporting form

## Data availability

All gene sequences, genomic, Pool-seq and RAD-Seq reads were deposited under the common project number PRJNA634624.

The following dataset was generated:

| Author(s) | Year | Dataset title | Dataset URL | Database and Identifier |
|---|---|---|---|---|
| Pan | 2020 | Sex determination in the Esociformes | https://www.ncbi.nlm.nih.gov/bioproject/PRJNA634624 | NCBI BioProject, PRJNA634624 |

The following previously published dataset was used:

| Author(s) | Year | Dataset title | Dataset URL | Database and Identifier |
|---|---|---|---|---|
| Rondeau EB, Minkley DR, Leong JS, Messmer AM, Jantzen JR, von Schalburg KR, Lemon C, Bird NH, Koop BF | 2014 | Esox lucius isolate CL-BC-CA-002, whole genome shotgun sequencing project | https://www.ncbi.nlm.nih.gov/nuccore/AZJR00000000.3/ | NCBI Nucleotide, GCA_000721915.2 |

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

## Appendix 1

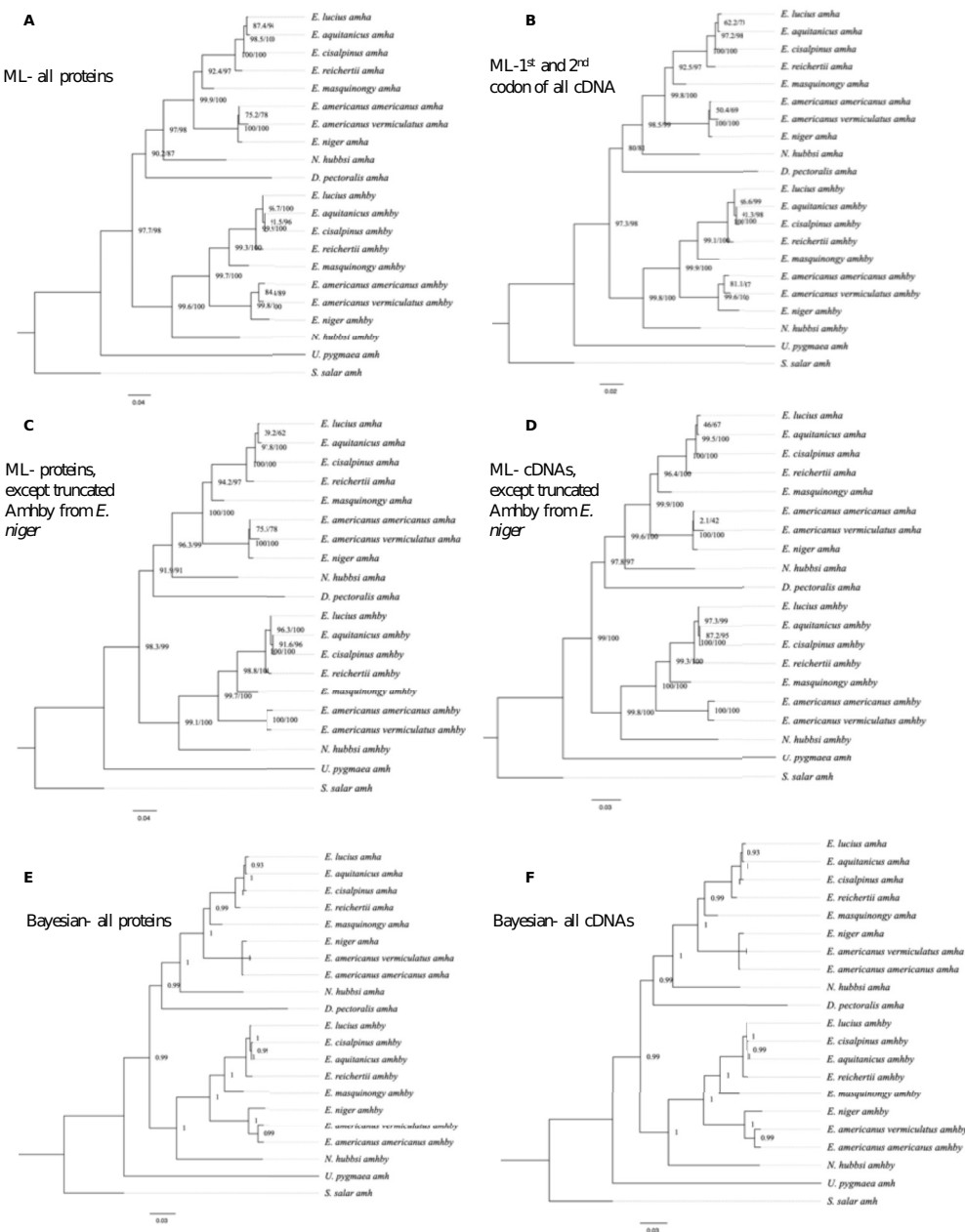

**Appendix 1—figure 1.** Additional phylogenetic reconstruction of *amha* and *amhby* orthologs from the *Esociformes* with amh sequence from *Salmo salar* as an outgroup. (**A**) Phylogenetic tree built by Maximum likelihood method implemented in IQ-TREE putative protein sequences of all identified *amh* homologs. (**B**) Phylogenetic tree built by Maximum likelihood method with only the 1st and 2nd codon of putative coding sequence of all identified *amh* homologs. (**C**) Phylogenetic tree built by Maximum likelihood with putative protein sequences of *amh* homologs without one highly truncated sequence from *E. niger*. (**D**) Phylogenetic trees built by Maximum likelihood with putative coding sequences of *amh* homologs without one highly truncated sequence from *E. niger*. (**E**) Phylogenetic tree built by Bayesian method implemented in PhyloBayes with putative protein sequences of all identified *amh* homologs. (**F**) Phylogenetic tree built by Bayesian method with putative coding sequences of all identified *amh* homologs. Bootstrap values are given on each nod of the tree and all trees are rooted with *amh* sequence from *S. salar*.

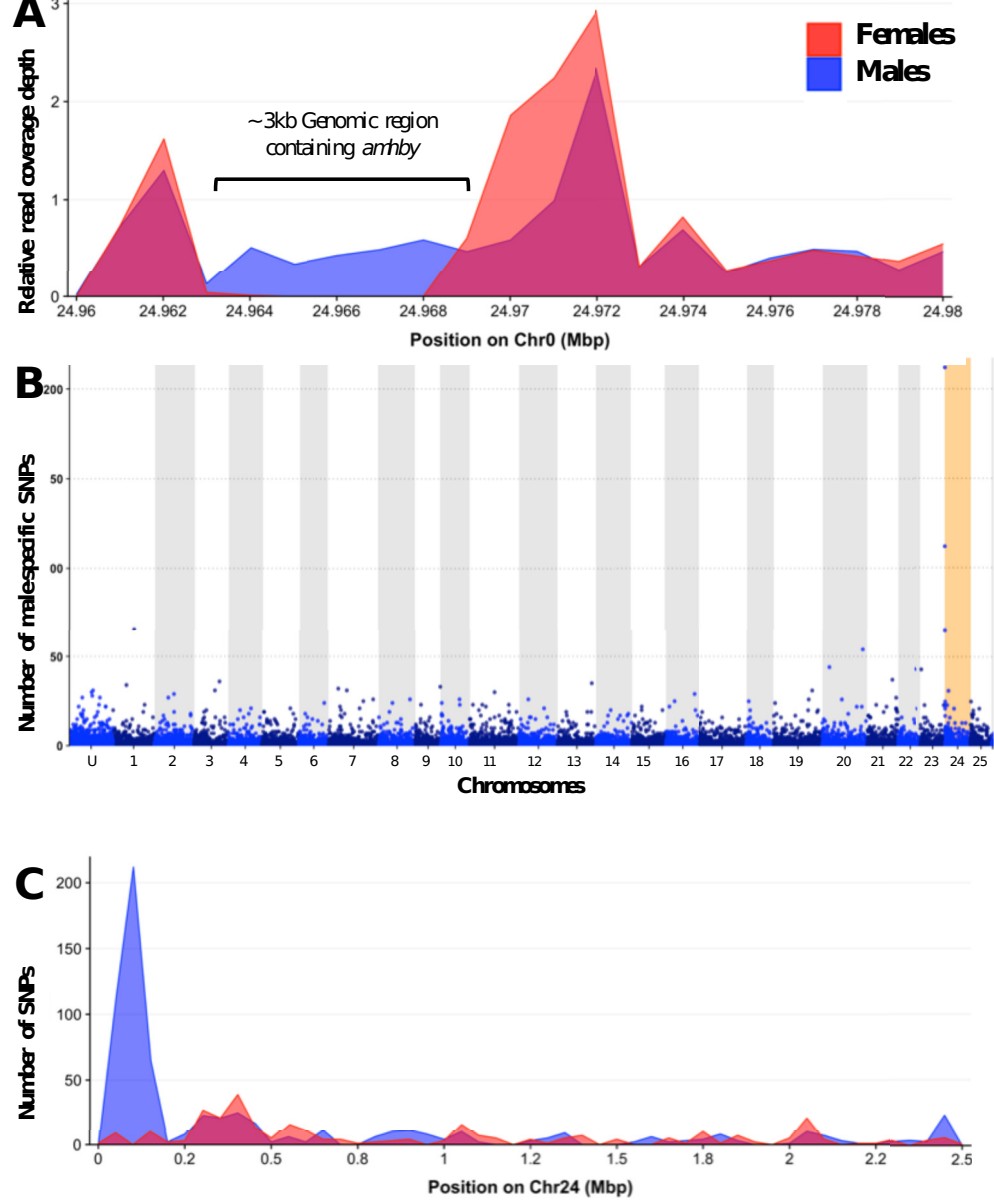

**Appendix 1—figure 2.** Analysis of sex differentiation in the genome of a male *E. masquinongy* with Pool-Seq data. (**A**) The relative to genome average coverage depth of male and female Pool-Seq reads on the region containing *amhby.* The male data are represented in blue and female in red. The 3 kb region containing *amhby* (unplaced_scaffold_RaGOO_chr0: 24,967,172–24,968,938) is covered by virtually no female reads and by male reads at a depth about half of the genome average depth. (**B**) Number of male-specific SNPs in 50 kb non-overlapping windows in each linkage group in the male genome of *E. masquinongy*. The window containing the highest number of male-specific SNPs on LG24 is highlighted. (**C**) Number of male and female-specific SNPs from Pool-Seq in 50 kb non-overlapping windows is plotted along LG. we found the highest number of male-specific SNPs (MSS) in a single window located around 100–150 kb on LG24 that contained 212 MSS and zero female-specific SNPs, while genome average was 1.57 MSS per 50 kb window (3.17 MSS per window when excluding 50 kb windows without MSS).This result indicates that the sex locus in

*Appendix 1—figure 2 continued*

*E. masquinongy* is homologous to the proximal end of LG24 in *E. lucius*. Besides LG24, no other linkage group contained a window enriched with MSS. Overall, this Pool-Seq analysis confirms the XX/XY SD system with a low differentiation between the X and Y chromosomes identified with the sex linkage and RAD-Seq analyses, as observed in *E. lucius* (*Pan et al., 2019*) and *N. hubbsi* (see above), and supports the hypothesis that the sex locus of *E. masquinongy* is similar to that of *E. lucius*.

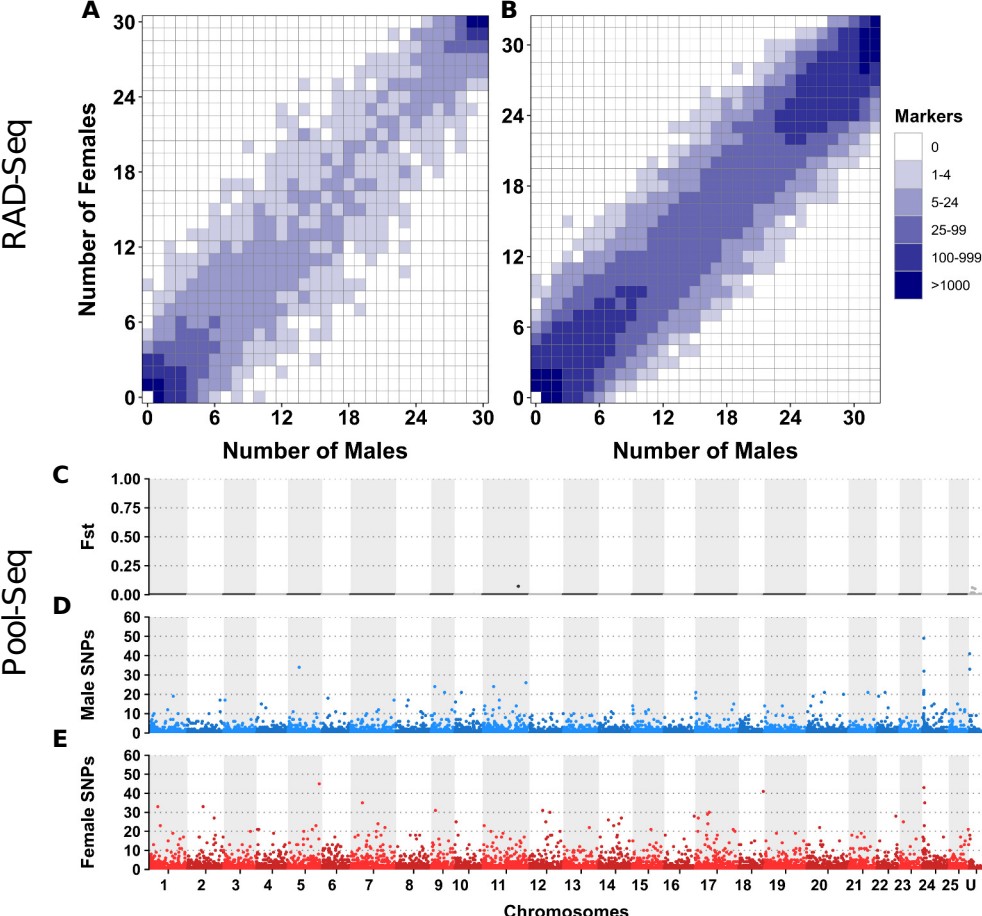

**Appendix 1—figure 3.** No differentiation between male and female genomes revealed by RAD-Seq and Pool-Seq in some North American populations of *E. lucius*. (**A-B**) Distribution of RADSex markers in males and females of two populations of *E. lucius* from Canada. The distribution of markers in males and females was computed with RADSex with a minimum depth of 5 to consider a marker as present in an individual for both datasets. In each tile plot, the number of males and number of females are represented on the horizontal and vertical axes respectively, and intensity of color of a tile indicates the number of markers present in the corresponding number of males and females. In both populations, we did not find any sex-linked markers among a total of 8,440,899 and 4,526,552 markers identified in each population, indicating that if there is a new sex locus in these populations its detection escapes the resolution of RAD-Seq (31.8 RAD markers per Mb on average) due to a very low differentiation between the new sex chromosome pair. (**C–E**) Analysis of male/female differentiation across Canadian *E. lucius* genome with Pool-Seq reads from 30 males and 30 females mapped to the reference genome (Accession number: GCA_004634155.1). Between sex $F_{ST}$ (**C**), female-specific SNPs (**D**) and male-specific SNPs (**E**) are computed for 50 kb non-overlapping

*Appendix 1—figure 3 continued on next page*

*Appendix 1—figure 3 continued*

windows across the 25 linkage groups (LGs) and unplaced scaffolds. We searched for 50 kb non-overlapping windows enriched with either male or female-specific SNP. Overall, the level of differentiation between males and females was low, and the highest Fst observed across the entire genome is 0.07 located between 0.397 Mb and 0.398 Mb on linkage group 11. Very low number of sex-specific SNPs were found for both male and female pools in this population, especially when comparing to the same analysis performed with data from an European population. In the European population, we found a genome average of 3.5 male-specific and 2.7 female-specific SNPs per 50 kb window, with peak windows containing 625 male-specific SNPs near the LG24 sex locus, and 217 female-specific SNPs at the proximal end of LG5. In comparison, the Canadian pools had roughly similar values of average genome sex-specific SNPs (3.3 male-specific and 2.9 female-specific SNPs per window), but with peak windows of sex-specific SNPs of 49 and 45 for males and females, respectively. Given this low amount of differentiation between the sexes, and that no particular chromosome was enriched with windows containing a high number of sex-specific SNPs, no region stood out to be the candidate region for the sex locus. Besides that, no 1 kb window with sex-specific read depth was found. Overall, no clear signal of a sex locus was observed across the entire genome supporting the hypothesis that the mainland USA and Canadian populations lack a well-differentiated sex determining region.

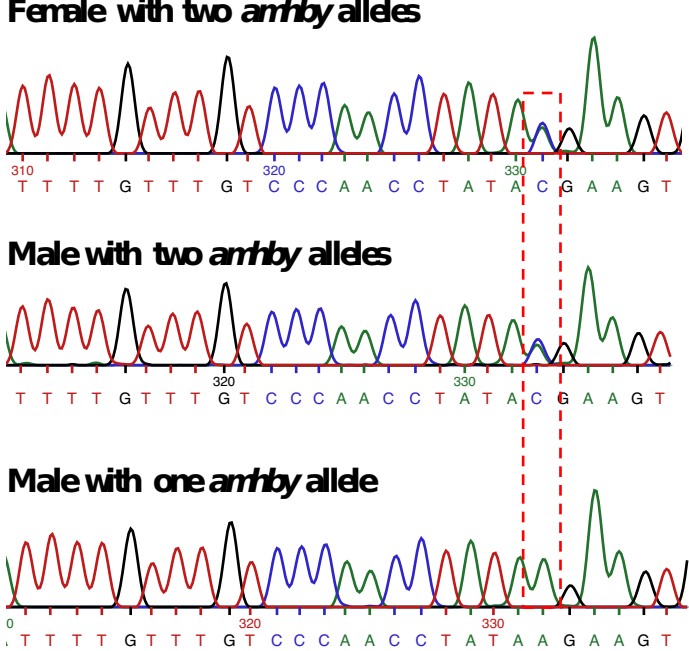

**Appendix 1—figure 4.** Sanger sequencing results for *E. masquinongy* one female and one male carrying two different alleles of *amhby* and one male carrying only one *amhby* allele. The base with a bi-allelic SNP is highlighted with the red dashed-line box.

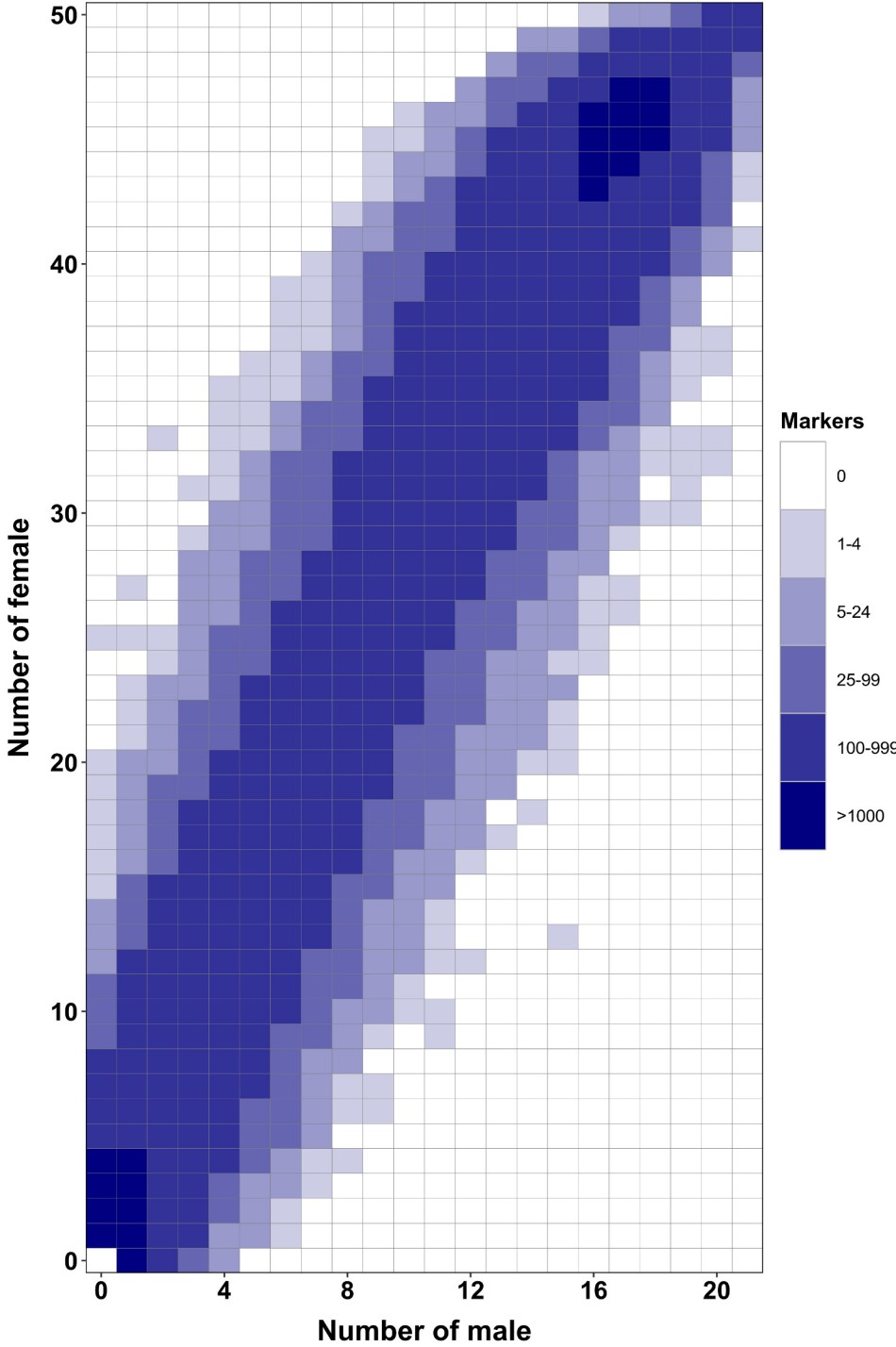

**Appendix 1—figure 5.** RAD-Seq analysis in a second *Esox masquinongy* population from Quebec (Canada) supports the lack of sex-specific marker in this population. Each tile plot shows the distribution of non-polymorphic RAD-Seq markers between phenotypic males (horizontal axis) and phenotypic females (vertical axis). The intensity of color for a tile corresponds to the number of markers present in the respective number of males and females. No tiles for which the association with phenotypic sex is significant (chi-square test, $p < 0.05$ after Bonferroni correction) were detected in this analysis. RAD-Seq reads data and samples information are found under NCBI bioproject PRJNA512459.

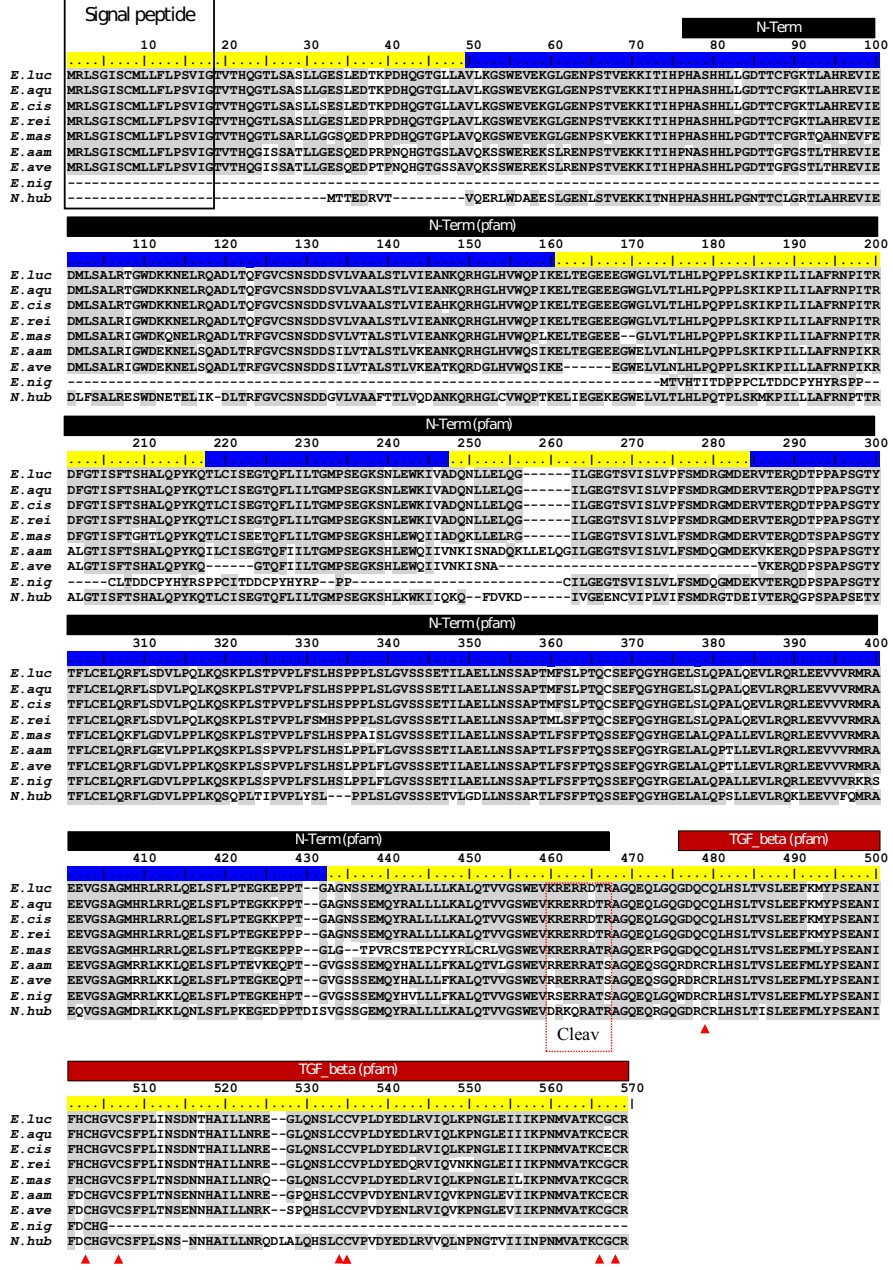

**Appendix 1—figure 6.** ClustalW alignment of Esocidae Amhby putative protein sequences. The signal peptide was predicted with SignalP (*Nielsen, 2017*). No signal peptide was detected in the Amhby sequences of both *E. niger* and *N. hubbsi*. N-terminal region (indicated by a black bar) and the Transforming growth factor beta like domain (indicated by a red bar) were predicted with the Motif Scan tool at ExPASy (*Gasteiger et al., 2003*) with the Pfam motif database (*Finn et al., 2014*) optimized for local alignments (pfam-fs, Pfam 32.0, September 2018). The seven exons of *amhby* are shown by the alternating blue and yellow colors on the sequence ruler. The seven conserved cysteines of the TGF-beta domain are indicated by red arrowheads. The region containing the putative Amh cleavage site (Cleav) is boxed in red. *E. luc* (*Esox lucius*), *E. aqu* (*E. aquitanicus*), *E. cis* (*E. cisalpinus*), *E. rei* (*E. reichertii*), *E. mas* (*E. masquinongy*), *E. aam* (*E. americanus americanus*), *E. ave* (*E. americanus vermiculatus*), *E. nig* (*E. niger*) and *N. hub* (*Novumbra hubbsi*).

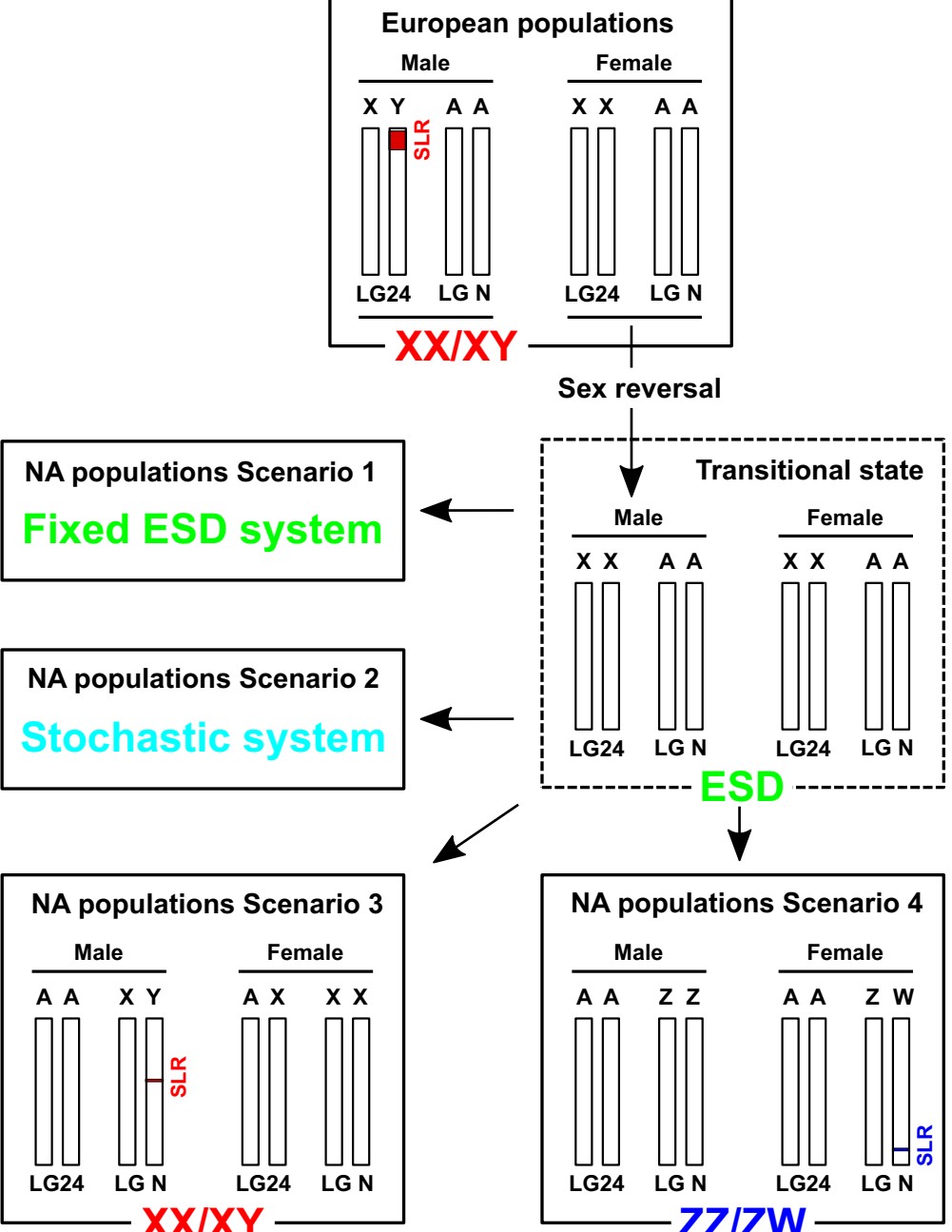

**Appendix 1—figure 7.** Hypothesized scenarios of sex determination system transition in North American populations of *E. lucius*. In the ancestral population, Linkage group 24 (LG24) was the sex chromosome (Y). Males were the heterogametic sex carrying one Y chromosome with a small (~300 kb) sex-locus region (SLR) containing *amhby* and one X chromosome without the SLR and *amhby*, while females carried two X chromosomes. During post-glaciation re-colonization of North America, the Y chromosome with *amhby* gene was not carried by individuals that made up the small populations that dispersed out of Alaska. These populations could still produce phenotypic males in the absence of a master sex determining gene via environment-induced sex reversal, a well-documented phenomenon among teleosts. In this transitional state, both males and females carry two X sex chromosomes (LG24). The current sex determination mechanism in these North American populations is still unknown: it could rely entirely on environmental sex determination (ESD), random sex determination, or new male (XY) or female (ZW) heterogametic genetic sex determination

*Appendix 1—figure 7 continued on next page*

*Appendix 1—figure 7 continued*

system with highly undifferentiated sex chromosomes. SLR: sex-locus region. ESD: environmental sex determination. NA: North America. LGN: un-specified Linkage group, that is ancestral autosomes (A) that are not LG24.

