## [Decision Letter]

**Acceptance summary:**

This paper investigates the evolution of a sex determination system in closely related species of Esociformes, focusing on the fate of a known master sex determining locus (SDL). By carefully tracing the evolution of this gene in an entire order of fishes, the work provides a more integrative picture of SDL evolution than previously available for any group. The authors suggest that genetic drift and in particular bottlenecks can facilitate loss of master SDLs in natural populations.

**Decision letter after peer review:**

Thank you for submitting your article "The rise and fall of the ancient northern pike master sex determining gene" for consideration by *eLife*. Your article has been reviewed by three peer reviewers, and the evaluation has been overseen by a Reviewing Editor and Detlef Weigel as the Senior Editor. The reviewers have opted to remain anonymous.

As you will see from the comments, the reviewers find the results of interest and would be happy to see them published. However, there was also unanimity that the writing requires considerable work: several points are imprecise or very difficult to parse and the structure is needlessly repetitive. We include individual comments below to guide you in what we expect will be fairly extensive revisions to the text.

Reviewer #2:

The manuscript presents the analysis of the evolution of the sex chromosomes across an entire teleost order. Specifically, the authors trace the evolution of a master sex determining gene (amhby), allowing to test recent hypothesis on how the birth of master sex determining (MSD) genes drive sex chromosome turnover in vertebrates. Data so far on evolution of sex-determination is based on distantly related clades, making it challenging to infer evolutionary pathways because data is obscured by large evolutionary times. Here the authors study closely related teleost species exhibiting sex determining (SD) system transitions, which allow to gain important insights on causes and mechanisms underlying SD turnover. Clearly, this is a topic that is a adapted to the broad readership of *eLife*.

The authors use draft genome assemblies and population genomic datasets in several species which allows them to investigate the evolution of *amhby* as a master SD gene across 65-90 MY. They propose that drift, exacerbated by bottleneck effects, can facilitate loss of MSD genes in wild populations. This is an interesting idea, that would provide a mechanism for the observed loss of sex chromosomes in some vertebrates. Overall the manuscript is well written and provides a wealth of interesting data and ideas. I believe the work in principle deserves publication in *eLife*. I have very few major comments, except that I find the Discussion a bit unstructured and lengthy and would be worth separating in sections. I would also like to see clear arguments supporting the XY -ZW system transition in *Dallia pectoralis*, I worry that just a few female markers from the RAD-seq are insufficient.

Reviewer #3:

Previous studies have found turnovers of sex-determining systems in fish, including Oryzias, tilapiine cichlids, salmonids, and sticklebacks, and this study uses another fish taxon, the order Esociformes, an old teleost group in which the *amhby* gene was known from previous studies by Pan's group to be the sex-determining gene. The study shows that, in 2 species, this gene does not behave as expected in all populations. The manuscript is well organized, but repeats things unnecessarily, and the only new information in the long Discussion is the statement about the *amhby* gene having "undergone an accelerated evolution", but I did not find the evidence for this described.

The manuscript suffers from creating the impression that these fish have Y chromosomes (which readers will understand to mean non-recombining chromosomes, or at least chromosomes with substantial non-recombining regions). This is not the case, and the Abstract and the text should not claim "Y-chromosome loss" or very recent loss of "the entire Y-chromosome". This suggests that there has been a Y-chromosome that became completely degenerated and was lost, creating an X0 sex-determining system. This is not the case, as the title is clear that this is not about loss of an entire Y chromosome. Instead, a single male-determining gene seems to characterize this fish group, and it has sometimes been lost, presumably in species that nevertheless have genetic sex-determination (though the text fails to make this clear). Overall, the study adds another example of changes in sex-determining genes in fish, so the results are less exciting than the text suggests,.

Confusion between Y chromosomes and sex linkage occurs in several parts of the text. For example, the start of subsection “Sex-linkage of *amhby* in the Esocidae lineage” should probably read simply "We tested whether the presence of *amhby* was strictly associated with the male phenotype, which indeed proved to be the case in populations of many of the species investigated in the genus *Esox* and *Novumbra*". As the associations were not significant in some cases, the text should not say "in all species". This association shows that *amhby* is probably the sex-determining gene in these species, but not that there is a non-recombining region that might be termed a Y-linked region or a Y chromosome.

Another related confusion is seen where the text mentions accumulation of deleterious mutations in the non-recombining sex-determining region. In the fish under study, however, there appear to be no non-recombining regions, so the reference to this is merely confusing to the reader.

The discussion of situations that favour changes in sex-determining genes could be replaced by citing a couple of reviews, as this study did not test any of the ideas, and is purely descriptive. In addition, this section speculates that genetic drift during a strong population bottleneck following postglacial dispersal might have facilitated the loss of the MSD gene (the text should not say "loss of the Y chromosome along with its MSD gene", as the chromosome is presumably not lost at all, and presumably the real meaning is that such an event might, for some reason have led to a turnover event). While a bottleneck may indeed be the reason for a change in the sex determining gene, and may have involved XX males due to sex reversal (as sometimes seen in captive *E. lucius*) after a colonization event that resulted in an all-female population, a single observation cannot be considered evidence in support of the hypothesis. Moreover, this suggestion is not a case of genetic drift, as stated, because it clearly involves selection favouring the ability of some XX genotypes to function as males.

Reviewer #4:

This paper presents interesting results from a family of fishes about the gain and loss of a master sex determining gene, one of the first studies of this kind. The conclusions are generally well justified and I would support publication after major revisions. The main problem is the writing, which has problems so distracting that at times it is hard to follow the point being made.

1) There are a large number of grammatical and stylistic errors. Two examples occur in the very first sentence of the Abstract:

• "Sex determination is an evolutionary highly dynamic process…." No, sex determination is a developmental process. The phrase "evolutionary highly dynamic process" is both factually wrong in many taxa (such as mammals) and grammatically incorrect.

• "… this dynamic evolution requires…" By definition, all evolution is dynamic.

It is very evident that different authors have written different sections. The Introduction and start of the Results are well-written, for example, but the other sections (e.g. the Discussion) are much weaker. The paper needs to be rewritten in a consistent style by a senior author with a command of the English language.

2) The organization of the paper is awkward and makes the story hard to follow. The two sections with big phylogenetic pictures ("Sex-linkage of *amhby* in the Esocidae lineage" and "Whole genome analyses of the evolution of sex determination") are interupted by sections that delve into two individual species (*Esox lucius* and *Esox masquinongy*). I don't know whether simply reordering the sections will fix the problem or whether some other organizational scheme would be best.

3) The Discussion is confusing. The Discussion paragraphs three and four are particularly rambling and disorganized. It's odd that ESD is not mentioned here.

[Editors' note: further revisions were suggested prior to acceptance, as described below.]

Thank you for submitting your article "The rise and fall of the ancient northern pike master sex determining gene" for consideration by *eLife*. Your article has been reviewed by three peer reviewers, and the evaluation has been overseen by a Reviewing Editor and Detlef Weigel as the Senior Editor. The following individual involved in review of your submission has agreed to reveal their identity: Mark Kirkpatrick (Reviewer #4).

The reviewers have discussed the reviews with one another and the Reviewing Editor has drafted this decision to help you prepare a revised submission.

This paper investigates the evolution of a sex determination system in closely related species of Esociformes, focusing on the rise and fall of a known master sex determining gene. By carefully tracing the evolution of this gene in an entire order of fish, it provides a bigger evolutionary picture than previously available for any group. The authors suggest that genetic drift and in particular bottlenecks can facilitate loss of master sex determination genes in natural populations.

Two of the reviewers appreciated the careful attention paid to their previous suggestions and concerns, and had no more comments, other than a minor tweak: "in subsection “Some populations of *Esox lucius* lost their Y chromosome and ancestral

master sex determining gene”, many Canadian readers will not be amused to see Quebec described as American." A third reviewer had more extensive suggestions about how the manuscript might be reframed; we leave it to your discretion to what extent you'd like to incorporate these in a revised version.

Reviewer #3:

Although the sentence should be revised to make the English correct, the conclusions described in the Abstract are potentially interesting: that (i) the sex-determining gene in the northern pike (*Esox lucius*) originated from a duplication at least 65 to 90 million years ago, and remained sex-linked for at least 56 million years in multiple related species without the chromosomes becoming differentiated into heteromorphic sex chromosome-like haplotypes, and that (ii) several independent species- or population-specific sex determination transitions have also occurred, including a recent loss of all genes in the Y-linked region of the *E. lucius* sex chromosome. However, the text does not have a clear conceptual focus on result (i), which is probably the most interesting result. I believe that long-term maintenance of small sex-linked regions has not previously been documented, although claims are made that cases exist, and such claims are plausible, in principle. Yet the manuscript appears not to mention previous suggestions, or explain that the case in pike may be the first adequately documented one. The lack of a conceptual focus should be corrected by a further revision. However, viewing this as the most interesting conclusion focuses attention on the evidence supporting it, I am not convinced that the conclusion is indeed supported. An alternative possibility is that the duplication arose long ago, and has independently, and more recently, become a sex-determining gene, and this is not excluded, as far as I can see. I highlight in my comments below some internal evidence that seems to support this alternative. The authors should be encouraged to think more carefully about this conclusion and whether the alternative can be excluded.

The Introduction, and much of the focus, is concentrated on result (ii) that turnovers appear to have occurred. However, turnovers are already well documented in many different taxa, and reporting new ones is less novel than conclusion (i). The Introduction has a lengthy section about the "limited option" hypothesis, which is almost untestable, as it has never seemed likely that the number of genes that could be involved in turnover events could be unlimited, but is much more likely that only a limited set of genes could have such functions. What is especially interesting about the evolution of sex-determination in fish such as the Esociformes is the possibility that, although many fish have young sex-linked regions that evolved by turnovers, some might have ancient ones. The Esociformes are stated to (i) have genetic sex- determination and (ii) to have diverged from the Salmoniformes (which are presumably the closest related group, though this is not stated explicitly) about 110 million years ago, making them of interest in relation to this question.

On this view, the main question is whether the evidence shows convincingly that the same gene as in *E. lucius* has remained the sex-determining in multiple related species, and still forms a small Y-linked region, as in *E. lucius*, where it is called amhby, and is a ~ 140 kb male-specific insertion of an autosomal gene, amha. If so, it is of interest to ask whether the chromosome region has become differentiated into a larger heteromorphic sex chromosome-like region, with a pair of haplotypes.

The relationships and taxonomy of the species are not well explained. The Introduction should explain that the two families (Esocidae and Umbridae) and 13 well-recognized species include the Esocidae species in the *Esox* and *Novumbra* genera studied (when these genus names are mentioned, we have not been told that they come from the Esocidae, and readers should not have to guess whether they are from the two separate families mentioned earlier, or the same one; it would be helpful to refer explicitly to the figure that shows the species phylogeny – I think it is Figure 1, but the figures are not labelled with their numbers). The important point readers need to understand at this place in the text is that all 7 *Esox* species and one *Novumbra* species surveyed (N. hubbsi) had two amh genes, whereas only one was found in two species, *Dalliapectoralis* and *Umbra pygmaea*, representing more basally diverging, outgroup species. This result suggests that *Esox* and *Novumbra* species could both have an *amhby* gene, as well as an amha one. Then we need to have the evidence explained that *Dallia pectoralis* and *Umbra pygmaea* have only the amha one (information is confusingly revealed in this and the next paragraph). It is misleading to write "all *Novumbra* species" unless more than one was surveyed, which Table 1 suggests was not the case. I also did not understand why the sentence says "However, two amh transcripts were readily identified in *E. lucius* (Pan et al., 2019), as well as in whole genome sequencing reads and assemblies (Supplementary file 1)". This sentence seems to confirm what was already stated above about *E. Lucius*, and perhaps it should be moved to an earlier place, saying "Moreover" rather than "However".

However, the next paragraph says that phylogenetic trees of the amh sequences showed "two well-supported clusters in all Esox, *Dallia* and *Novumbra* species", suggesting *Dallia* has both copies. Some conclusions about these species are based on limited information in transcriptome databases, and some on other information, and the evidence should be more clearly and coherently explained.

At this point, if I have guessed the correct meaning, I think that all *Esox* species, plus one species each of the genera *Novumbra* (but not *Dallia* and Umbra), have sequences that cluster with the *E. lucius* amha one, and are likely to be orthologs of this gene, and also sequences that cluster with the *E. luciusamhby* one, suggesting that the amh gene duplication preceded the split between the genera *Esox* and *Novumbra*, but perhaps occurred after the split from *U. pygmaea* (and perhaps *Dallia*). The text says that the "duplication happened after the divergence of Esocidae and Umbridae lineages", which suggests that *Novumbra* (with both copies) is not in the Umbridae. Please can the relationships be made clear before the results from them are described.

Clearly, the important questions are now whether the *amhby* copies are sex-determining genes in all species that carry the duplicate copy, and whether they acquired this function in the common ancestor of *Esox* and *Novumbra*, and retained this ever since. To test this, the study tested whether *amhby* was found in all males and absent from all females, in species additional to the *E. lucius* sample previously studied.

The tests indeed detected associations between the male phenotype and the presence of *amhby* in most species of *Esox* (Table 1), but the associations were absent or weak in some *Esox* samples, including some North American *E. lucius* populations, and incomplete in the *Novumbra* species sample.

Overall, it is clear that the duplication is ancient, and pre-dates the split between *Esox* and *Novumbra*, but there does not seem to be strong evidence for the conclusion that it has maintained a male-determining function since the duplication occurred. The alternative cannot be excluded that the duplicate copy has acquired a sex-determining function independently in *Esox* and *Novumbra*, as well as losing amhb several times. In fact, this study seems a nice example of how much evidence is needed before one can be sure that a sex-determining gene has been maintained for a long time.

The claim that the same region has remained as an undifferentiated sex-determining or Y-linked region for a long evolutionary time comes from the Pool-Seq analysis of *E. masquinongy*, one of the species with an incomplete association between the presence of a copy of the amh duplicate, to compare its size and location (as a putative sex locus) with that of *E. lucius*. Male-specific variants (described rather vaguely as "sex-specific heterozygosity") were detected in just a single genome region of less than 50 kb, and the region is homologous to the proximal end of the *E. lucius* LG24, where its sex-determining locus is located. This indeed suggests that both *E. lucius* and *E. masquinongy* have a physically small male-determining locus at the same location. However, it appears to have a weakened or partially lost function in *E. masquinongy*.

However, the results do not tell us that they the male-determining function is ancestral, rather than independently evolved, in *Esox* and *Novumbra*. Indeed, subsection “Evolution of the structure of the *amhby* gene in the Esocidae” reports information that may suggest independent evolution in E. niger (in which amhby is strongly associated with maleness) and N. hubbsi. In both, the predicted Amhby protein is truncated in regions known to be important for this gene's function, but in E. niger the truncation is in its C-terminal part, whereas it affects the N-terminal part of the N. hubbsi sequence, and exon 1 encodes only eight amino acids, with no homology to the amino acid-sequence in other Esocidae. It is not explained whether the *E. lucius* amhby copy is complete, or how it differs from amha. On the interpretation that these changes evolved independently, the similarity of other parts of the sequence (which provides the signal in the phylogeny) is potentially rather misleading.

The manuscript also reports evidence that sex-determining systems evolved independently in the 2 species that do not have the amh duplicate, *D. pectoralis* and *U. pygmaea*.

It then turns to plausible suggestions about why *Esox* species might lose their male-determining factor. This section is rather unorganised, and should be greatly shortened to make clear what hypotheses were tested, and what the results were. Loss of genetic sex-determination is plausible in colonizing or low density situations, and the findings should be related to this concept.

---

## [Author Response]

Reviewer #2:The manuscript presents the analysis of the evolution of the sex chromosomes across an entire teleost order. Specifically, the authors trace the evolution of a master sex determining gene (amhby), allowing to test recent hypothesis on how the birth of master sex determining (MSD) genes drive sex chromosome turnover in vertebrates. Data so far on evolution of sex-determination is based on distantly related clades, making it challenging to infer evolutionary pathways because data is obscured by large evolutionary times. Here the authors study closely related teleost species exhibiting sex determining (SD) system transitions, which allow to gain important insights on causes and mechanisms underlying SD turnover. Clearly, this is a topic that is a adapted to the broad readership of eLife.The authors use draft genome assemblies and population genomic datasets in several species which allows them to investigate the evolution of amhby as a master SD gene across 65-90 MY. They propose that drift, exacerbated by bottleneck effects, can facilitate loss of MSD genes in wild populations. This is an interesting idea, that would provide a mechanism for the observed loss of sex chromosomes in some vertebrates. Overall the manuscript is well written and provides a wealth of interesting data and ideas. I believe the work in principle deserves publication in eLife. I have very few major comments, except that I find the Discussion a bit unstructured and lengthy and would be worth separating in sections. I would also like to see clear arguments supporting the XY -ZW system transition in Dallia pectoralis, I worry that just a few female markers from the RAD-seq are insufficient.

The manuscript has been deeply revised to incorporate supplementary notes that are not allowed by the *eLife* format and also indeed to take into consideration reviewers' comments. The Discussion has been shortened and organized.

Concerning the *Dallia pectoralis* sex determination system, we now provide results from an independent Pool-Seq analysis supporting the existence of female-specific sequences in the Dallia genome. Results are in agreement with the ZZ/ZW SD system revealed by the RAD-seq analysis. We answered this comment in more detail below.

Reviewer #3:Previous studies have found turnovers of sex-determining systems in fish, including Oryzias, tilapiine cichlids, salmonids, and sticklebacks, and this study uses another fish taxon, the order Esociformes, an old teleost group in which the amhby gene was known from previous studies by Pan's group to be the sex-determining gene. The study shows that, in 2 species, this gene does not behave as expected in all populations. The manuscript is well organized, but repeats things unnecessarily, and the only new information in the long Discussion is the statement about the amhby gene having "undergone an accelerated evolution", but I did not find the evidence for this described.

As mentioned above, the manuscript has been thoroughly revised to take into consideration all reviewers' comments and to incorporate our previous supplementary notes that are not allowed by the *eLife* format. The Results section has been changed to incorporate some information that was previously in the supplementary notes and the Discussion has been greatly shortened.

Regarding the statement “undergone an accelerated evolution”, we agree that it is not a result we have support for, and we then thus changed our sentence to “ The northern pike MSD gene *amhby* substantially diverges from its autosomal paralog *amha*, suggesting an ancient origin (Pan et al., 2019), unlike other cases of amh duplication in fishes which appear to be comparatively young (Hattori et al., 2013; Li et al., 2015)”. We attribute the high level of sequence divergence between *amhby* and its autosomal paralog *amha* to ancient origin rather than to accelerated evolution.

The manuscript suffers from creating the impression that these fish have Y chromosomes (which readers will understand to mean non-recombining chromosomes, or at least chromosomes with substantial non-recombining regions). This is not the case, and the Abstract and the text should not claim "Y-chromosome loss" or very recent loss of "the entire Y-chromosome". This suggests that there has been a Y-chromosome that became completely degenerated and was lost, creating an X0 sex-determining system. This is not the case, as the title is clear that this is not about loss of an entire Y chromosome. Instead, a single male-determining gene seems to characterize this fish group, and it has sometimes been lost, presumably in species that nevertheless have genetic sex-determination (though the text fails to make this clear). Overall, the study adds another example of changes in sex-determining genes in fish, so the results are less exciting than the text suggests,.

The reviewer’s comment revolves around the definition of a sex chromosome. A sex chromosome is generally defined as a chromosome that contains a major sex determining genetic factor that controls whether an individual becomes a male or female. It does not mean that one of the sex chromosome pairs has to be nearly devoid of genes that do not directly control sex, even though this is true of the mammalian Y chromosome. In our previous study, besides the identification and functional demonstration of *amhby* as the MSD gene in northern pike, we also identified a 300kb male specific region on a specific chromosome and demonstrated an absence of recombination in male around this region with linkage mapping. According to the generally accepted definition of a sex chromosome, this then would be a Y chromosome even though it shares a substantial number of genes with the X chromosome. Although this species doesn’t not have cytologically heteromorphic sex chromosomes, it thus has a Y-chromosome with a small, yet clearly Y-specific region. In this current study, when we say the loss of the Y-chromosome, we are not implying that it was replaced by a X-O system, but rather that all the 300kb of Y-specific sequences are absent in the North American populations of northern pike. This loss is not just limited to a single male-determining gene but to the entire, albeit small, Y-chromosome-specific sequence (inserted sequence on Y chromosome, as well as differentiated regions from the X chromosome). We have added Appendix 1—figure 7 to clarify this point. There is little divergence time (~8000 years) between the population with *amhby* (Alaska populations) and without amhby (Canada and mainland USA). Considering this, we suggest that this loss of all Y-specific sequences could have been caused by having all founders in a bottle-necked population being genetic females (XX individuals), some of which were sex-reversed phenotypic males, during postglacial dispersal. Hence, no ancestral Y-chromosomes remained in these populations that re-colonized North America out of Alaska. We agree that it is probably another case of change in sex-determining gene in fish, however, we think that the potential mechanism underlying this change has not been described before as it puts forward a non-adaptive hypothesis for such a transition and highlights the importance of considering population demography in understanding the drivers for transitions in sex determination systems.

We realize, however, that we probably didn’t describe clearly enough our previous finding on the Y chromosome of *E. lucius* (Pan et al., 2019); thus, we added additional information in the Introduction : “Previously, we demonstrated that a male-specific duplication of the anti-Müllerian hormone gene (*amhby*) is the MSD gene in Esociformes, and that this gene is located in a small, ~ 140 kb male-specific insertion on the Y chromosome of northern pike (*Esox lucius*) (Pan et al., 2019).”

Confusion between Y chromosomes and sex linkage occurs in several parts of the text. For example, the start of subsection “Sex-linkage of amhby in the Esocidae lineage” should probably read simply "We tested whether the presence of amhby was strictly associated with the male phenotype, which indeed proved to be the case in populations of many of the species investigated in the genus Esox and Novumbra". As the associations were not significant in some cases, the text should not say "in all species". This association shows that amhby is probably the sex-determining gene in these species, but not that there is a non-recombining region that might be termed a Y-linked region or a Y chromosome.

We agree we cannot say that a “significant association” is found in all species given the small sample size for the two species where we only had access to a limited number of museum samples. We have followed the reviewer’s suggestion and updated the text to “A significant association between male phenotype and the presence of *amhby* was found in most species investigated in the genus *Esox* and *Novumbra*”.

But we are confused over the reviewer’s view on the relationship between master sex determining genes and sex chromosomes. From both classic literature (Bull, 1983; Charlesworth et al., 2005) to more recent reviews on sex chromosome formation and evolution (Vicoso, 2019; Wright et al., 2016), new sex chromosomes arise after the acquisition of a new master sex-determining gene (if brought by a duplication / insertion event) or a new master sex determining allele (if brought by an allelic diversification process). From there, this new sex locus on a new sex chromosome can either become heteromorphic by extension of the non-recombining region around the original sex locus, or it might remain in its current state with as little difference from the X chromosome as one SNP on the sex-determining gene as found in *Takifugu* (Ieda et al., 2018; Kamiya et al., 2012). But whatever their differentiation level, both at the molecular and karyotype levels, a chromosome containing a sex locus is (always) considered as a sex chromosome. So, unless we missed something important, or we misunderstood the reviewer’s comment, we decided to stick with the idea that the presence of a master sex determination gene in one sex indirectly informs about sex determination systems and sex-chromosomes.

Another related confusion is seen where the text mentions accumulation of deleterious mutations in the non-recombining sex-determining region. In the fish under study, however, there appear to be no non-recombining regions, so the reference to this is merely confusing to the reader.

We agree that we only have evidence for that in the European populations of Northern pike (Pan et al., 2019), from our linkage map that is showing that recombination is suppressed around the 300kb of the Y-specific region on the proximal end of LG24. We agree that we do not know precisely the size of the sex locus in many of the species investigated, except for a very rough estimation based on number of RAD-Seq markers. However, these sequence differences between male and female genomes result from reduced recombination around the sex determining locus. If X and Y chromosomes are free to recombine in these fish, fixed differences between the male and female genomes would involve solely the few nucleotides specifically within the MSD gene itself. Nevertheless, to help readers, we updated our text to use either “regions of X-Y differentiation” or “Y-chromosome-specific regions” instead of non-recombining regions.

The discussion of situations that favour changes in sex-determining genes could be replaced by citing a couple of reviews, as this study did not test any of the ideas, and is purely descriptive.

We have followed the reviewer’s suggestion and shortened significantly discussions of our hypothesis of how the transitions could have happened.

In addition, this section speculates that genetic drift during a strong population bottleneck following postglacial dispersal might have facilitated the loss of the MSD gene (the text should not say "loss of the Y chromosome along with its MSD gene", as the chromosome is presumably not lost at all, and presumably the real meaning is that such an event might, for some reason have led to a turnover event). While a bottleneck may indeed be the reason for a change in the sex determining gene, and may have involved XX males due to sex reversal (as sometimes seen in captive E. lucius) after a colonization event that resulted in an all-female population, a single observation cannot be considered evidence in support of the hypothesis. Moreover, this suggestion is not a case of genetic drift, as stated, because it clearly involves selection favouring the ability of some XX genotypes to function as males.

Again, we have shown that this loss is not restricted to just to the MSD gene, but to the entire sex locus with at least 140kb of Y-chromosome specific sequences. Indeed, the former X chromosome (=LG24) is still present in the population and we never suggested that the X chromosome is in a hemizygous state in these populations. We changed large parts of our Discussion and we hope that it is more clearly stated now that it is not a transition from an XY to XO system but a rapid disappearance of the ancestral Y chromosome in some of the Northern American populations we survey. To clarify this point, we have added a diagram to better illustrate our hypothesis that can be found in Appendix 1—figure 7.

Regarding whether there is selection favoring the ability of some XX genotypes to function as males, we think this would be a process independent from the loss of the Y chromosome. We didn’t exclude the involvement of selection in generating a potential new mechanism for sex determination, but we are suggesting that the loss of the Y-chromosome was not due to selection (for instance, the “hot-potato” model of sex-chromosome turnover (Blaser et al., 2013) but to a sudden drift occurring in a bottlenecked population with environmental or stochastic sex reversal.

Reviewer #4:This paper presents interesting results from a family of fishes about the gain and loss of a master sex determining gene, one of the first studies of this kind. The conclusions are generally well justified and I would support publication after major revisions. The main problem is the writing, which has problems so distracting that at times it is hard to follow the point being made.

As mentioned above, the manuscript has been thoroughly revised to take into consideration all reviewers' comments and to incorporate our previous supplementary notes that are not allowed by the *eLife* format. The Results section has been changed to incorporate some information that were previously in the supplementary notes and the Discussion has been greatly shortened.

1) There are a large number of grammatical and stylistic errors. Two examples occur in the very first sentence of the Abstract:• "Sex determination is an evolutionary highly dynamic process…." No, sex determination is a developmental process. The phrase "evolutionary highly dynamic process" is both factually wrong in many taxa (such as mammals) and grammatically incorrect.• "… this dynamic evolution requires…" By definition, all evolution is dynamic.It is very evident that different authors have written different sections. The Introduction and start of the Results are well-written, for example, but the other sections (e.g. the Discussion) are much weaker. The paper needs to be rewritten in a consistent style by a senior author with a command of the English language.

Thanks for pointing out these mistakes in the Abstract. We have now changed the first sentence to “The evolution of sex determination mechanisms varies across taxa, and an understanding of this variation requires comparative studies among closely related species.”

The whole manuscript has been edited for style and grammar by native English-speaking senior co-authors.

2) The organization of the paper is awkward and makes the story hard to follow. The two sections with big phylogenetic pictures ("Sex-linkage of amhby in the Esocidae lineage" and "Whole genome analyses of the evolution of sex determination") are interupted by sections that delve into two individual species (Esox lucius and Esox masquinongy). I don't know whether simply reordering the sections will fix the problem or whether some other organizational scheme would be best.

We changed the ordering of these sections according to the reviewer’s suggestion. The placing of “Whole genome analyses of the evolution of sex determination" right after "Sex-linkage of *amhby* in the Esocidae lineage" was our original ordering of the sections and we agree it is probably more fluid to give the big picture before venturing into the population level of SD variation it the two species.

3) The Discussion is confusing. The Discussion paragraphs three and four are particularly rambling and disorganized. It's odd that ESD is not mentioned here.

We agree that our original Discussion was too long and not clearly structured. We have since substantially shortened the Discussion, especially for the paragraphs pointed out by the reviewer. We actually mentioned ESD, but it was probably buried by our “rambling”. The discussion on ESD can now be found in Discussion paragraph six.

[Editors' note: further revisions were suggested prior to acceptance, as described below.]

This paper investigates the evolution of a sex determination system in closely related species of Esociformes, focusing on the rise and fall of a known master sex determining gene. By carefully tracing the evolution of this gene in an entire order of fish, it provides a bigger evolutionary picture than previously available for any group. The authors suggest that genetic drift and in particular bottlenecks can facilitate loss of master sex determination genes in natural populations.Two of the reviewers appreciated the careful attention paid to their previous suggestions and concerns, and had no more comments, other than a minor tweak: "in subsection “Some populations of Esox lucius lost their Y chromosome and ancestralmaster sex determining gene”, many Canadian readers will not be amused to see Quebec described as American."

Thanks for this positive feedback on our revised manuscript. We have changed the text to read: “an American population that lacks *amhby* (Quebec, Canada)” to “ a North American population that lacks *amhby* (Quebec, Canada)” to make sure that Quebec is not to be confused as a part of America (USA), but the continent of North America.

A third reviewer had more extensive suggestions about how the manuscript might be reframed; we leave it to your discretion to what extent you'd like to incorporate these in a revised version.

We carefully read reviewer #3 comments and all his(her) “extensive suggestions”, and we tried to provide answers and to take into consideration his(her) suggestions to the largest possible extent without totally rewriting the manuscript

Reviewer #3:Although the sentence should be revised to make the English correct,

We agree that some Abstract sentences were not easy to understand. This was due to last minutes changes to fit with the tight word count required for the Abstract during submission. We slightly rewrote the Abstract to clarify these sentences. This new Abstract now reads as:

“The understanding of the evolution of variable sex determination mechanisms across taxa requires comparative studies among closely related species. Following the fate of a known master sex-determining gene, we traced the evolution of sex determination in an entire teleost order (Esociformes). We discovered that the northern pike (*Esox lucius*) master sex-determining gene originated from a 65 to 90 million-year-old gene duplication event and that it remained sex-linked on undifferentiated sex chromosomes for at least 56 million years in multiple species. We identified several independent species- or population-specific sex determination transitions, including a recent loss of a Y-chromosome. These findings highlight the diversity of evolutionary fates of master sex-determining genes and the importance of population demographic history in sex determination studies. We hypothesize that occasional sex reversals and genetic bottlenecks provide a non-adaptive explanation for sex determination transitions.”

the conclusions described in the Abstract are potentially interesting: that (i) the sex-determining gene in the northern pike (Esox lucius) originated from a duplication at least 65 to 90 million years ago, and remained sex-linked for at least 56 million years in multiple related species without the chromosomes becoming differentiated into heteromorphic sex chromosome-like haplotypes, and that (ii) several independent species- or population-specific sex determination transitions have also occurred, including a recent loss of all genes in the Y-linked region of the E. lucius sex chromosome. However, the text does not have a clear conceptual focus on result (i), which is probably the most interesting result.

We understand that some colleagues, like reviewer #3, will find point (i) more interesting than point (ii) and may be frustrated by the fact we did not develop further the discussion on this point. However, we do not think that point (i) should deserve more attention than point (ii), and our current manuscript currently provides a balanced discussion on these two points. As this prioritization is purely subjective we would prefer not changing our manuscript with regards to this comment.

I believe that long-term maintenance of small sex-linked regions has not previously been documented, although claims are made that cases exist, and such claims are plausible, in principle. Yet the manuscript appears not to mention previous suggestions, or explain that the case in pike may be the first adequately documented one.

To our knowledge there are a few known cases of long-term maintenance of small sex-linked regions in fish that we actually discussed in our manuscript like for instance in some *Takifugu* species in which a very small sex locus (restricted to a single causal SNP in the *amhr2* gene, probably the smallest sex locus described in any species) has been conserved for more than 10 million years. See our Discussion “In line with our present results, substantially undifferentiated sex chromosomes have also been maintained over relatively long evolutionary periods in some *Takifugu* fish species (Kamiya et al., 2012) “. We then do not think that we can claim that the pike case is the first documented one.

The lack of a conceptual focus should be corrected by a further revision. However, viewing this as the most interesting conclusion focuses attention on the evidence supporting it, I am not convinced that the conclusion is indeed supported.

With respect to “long-term maintenance of small sex-linked regions” what we found is multiple species with a conserved MSD gene (*amhby*) and a small (or undifferentiated) sex locus. But we didn’t claim that it is a conserved ancestral locus as we only provided evidence that homologous regions containing *amhby* have been used as the sex locus in two species that diverged 45 million year ago i.e., *E. lucius* and *E. masquinongy*. As we don’t have a chromosome level genome assembly for *E. masquinongy*, we chose not to make inferences on the age of the sex chromosome or this shared sex locus. In other species we only found a conservation of the MSD gene, but this does not necessarily means that such a conserved ancestral MSD gene is located in a conserved sex locus and /or sex chromosome. For instance, in salmonids, the MSD gene sdY has been conserved over 50 millions year, but the linkage group hosting this MSD gene is different in many species.

An alternative possibility is that the duplication arose long ago, and has independently, and more recently, become a sex-determining gene, and this is not excluded, as far as I can see. I highlight in my comments below some internal evidence that seems to support this alternative. The authors should be encouraged to think more carefully about this conclusion and whether the alternative can be excluded.

We agree that in theory, *amhby* could have been recruited independently as a MSD gene long after the divergence of *Esox* and *Novumbra*. However, the hypothesis that *amhby* is the ancestral MSD is more parsimonious than *amhby* gaining function, independently and much more later, in the *Esox* and *Novumbra* lineage. We now discuss this point in the main text, which reads “Although we cannot rule out the possibility that *amhby* was recruited independently as the MSD gene in *Esox* and in *Novumbra,* our results suggest the more parsimonious hypothesis that *amhby* likely acquired an MSD function before the diversification of Esocidae at least 56 Mya.“

The Introduction, and much of the focus, is concentrated on result (ii) that turnovers appear to have occurred. However, turnovers are already well documented in many different taxa, and reporting new ones is less novel than conclusion (i).

As mentioned above, whether point (i) or point (ii) is more “novel” is purely subjective and we would prefer not changing our manuscript with regards to this point (i) prioritization. We also disagree with reviewer #3 idea that “turnovers are already well documented in many different taxa” and in our Introduction we wrote that “Although comparative studies have been accomplished in medakas, poeciliids, tilapiine cichlids, salmonids, and sticklebacks (Kikuchi and Hamaguchi, 2013), transitions of SD systems in relation to the fate of known MSD genes within closely related species have only been explored in medakas (Myosho et al., 2015) and salmonids (Guiguen et al., 2018).”. This clearly states that “turnovers” (SD transitions) have not been clearly documented in many teleost groups, and that outside, medakas and salmonids, most of these transitions were documented without any knowledge on the identity and the fate of the MSD gene.

The Introduction has a lengthy section about the "limited option" hypothesis, which is almost untestable, as it has never seemed likely that the number of genes that could be involved in turnover events could be unlimited, but is much more likely that only a limited set of genes could have such functions.

We agree with reviewer #3 that the “limited option” hypothesis is not a testable hypothesis but more a conceptual idea. We then changed the wording to “limited option” idea. However, as far as MSD gene turnovers are concerned, the list of potential candidates maybe bigger than reviewer #3 expectation (if his(her) expectation is that all genes with a key function in the sex differentiation network could be selected as a new MSD gene), as unexpected genes like the sdY MSD gene of salmonids can also evolve to become new MSD genes. Finding these unusual MSD genes may be just more difficult, leading to a hidden complexity that could actually challenge this “limited option” idea.

Concerning our lengthy section on the "limited option" hypothesis it is just a single sentence “These teleost MSD genes also provided empirical support for the “limited option” hypothesis that states that certain genes known to be implicated in sex differentiation pathways are more likely to be recruited as new MSD genes”, that is used to introduce the following idea that “The majority of these recently discovered MSD genes, however, were phylogenetically scattered, making it challenging to infer evolutionary patterns and conserved themes of sex chromosomes and / or MSD gene turnovers”. We then do not think that this is a particularly lengthy section and we would like to keep our text as it was.

What is especially interesting about the evolution of sex-determination in fish such as the Esociformes is the possibility that, although many fish have young sex-linked regions that evolved by turnovers, some might have ancient ones. The Esociformes are stated to (i) have genetic sex- determination and (ii) to have diverged from the Salmoniformes (which are presumably the closest related group, though this is not stated explicitly) about 110 million years ago, making them of interest in relation to this question.

We indeed agree with reviewer #3 that the evolution of sex determination is Esociformes is an interesting question. But we cannot introduce our manuscript by saying that Esociformes are known to have a genetic sex determination mechanism as this was not known before our present manuscript (with the exception of northern pike). We also think that the fact that salmoniformes are the closest order to Esociformes is not the main point that drove our study (may be just interesting in light of our results as they do not share the same MSD gene). Based on that we feel that we do not need to change this introduction part.

We however slightly changed one sentence of our Introduction to clearly mention that salmoniformes are the closest order to Esociformes in line with reviewer #3 suggestion. This sentence now reads as “Esociformes is a small order of teleost fishes (Figure 1) that diverged from their sister group Salmoniformes about 110 million years ago (Mya) and diversified from a common ancestor around 90 Mya (Campbell et al., 2013; Campbell and Lopéz, 2014) “.

On this view, the main question is whether the evidence shows convincingly that the same gene as in E. lucius has remained the sex-determining in multiple related species, and still forms a small Y-linked region, as in E. lucius, where it is called amhby, and is a ~ 140 kb male-specific insertion of an autosomal gene, amha. If so, it is of interest to ask whether the chromosome region has become differentiated into a larger heteromorphic sex chromosome-like region, with a pair of haplotypes.

We agree that the question on how an initial small sex locus on a sex chromosome can differentiate into a larger sex locus is indeed a very interesting question. However, the *E. lucius* sex locus (that we investigated more extensively in our previous paper (Pan et al. 2019) with both a linkage map to delineate the region of suppressed recombination and population data to explore X/Y differentiation) is probably not large enough to consider its sex chromosomes as heteromorphic at least if we follow a currently well accepted definition like the one given by Wright et al., 2016 (10.1038/ncomms12087) in which homomorphic sex chromosomes were defined as “sex-chromosomes that exhibit few differences from each other in size and gene content, and are difficult or impossible to distinguish from karyotype data alone”.

The relationships and taxonomy of the species are not well explained. The Introduction should explain that the two families (Esocidae and Umbridae) and 13 well-recognized species include the Esocidae species in the Esox and Novumbra genera studied (when these genus names are mentioned, we have not been told that they come from the Esocidae, and readers should not have to guess whether they are from the two separate families mentioned earlier, or the same one; it would be helpful to refer explicitly to the figure that shows the species phylogeny – I think it is Figure 1, but the figures are not labelled with their numbers).

We anticipated this taxonomy and species phylogeny difficulties for non (fish and pike) specialist readers and because of that we have dedicated the first main figure of the manuscript for this purpose. But to make it even more clear we now mention that both the *Esox* and *Novumbra* genera belong to the Esocidae. “With two families Esocidae, including *Esox*, *Novumbra* and *Dallia* and Umbridae, including *Umbra*, and 13 well-recognized species (Warren et al., 2020)”.

We however, do not understand why figures were not labelled with their corresponding numbers in the final pdf provided for reviewing as we clearly filled that information in the *eLife* submission website. We apologize for this inconvenience but it is clearly not our fault.

The important point readers need to understand at this place in the text is that all 7 Esox species and one Novumbra species surveyed (N. hubbsi) had two amh genes, whereas only one was found in two species, Dallia pectoralis and Umbra pygmaea, representing more basally diverging, outgroup species.

We agree and this is exactly what is stated in our manuscript i.e., “We found two *amh* genes in all surveyed *Esox* and *Novumbra* species. In the more basally diverging species, *Dallia pectoralis* and *Umbra pygmaea*, we found only one *amh* gene …”.

This result suggests that Esox and Novumbra species could both have an amhby gene, as well as an amha one. Then we need to have the evidence explained that Dallia pectoralis and Umbra pygmaea have only the amha one (information is confusingly revealed in this and the next paragraph).

We understand that this may be difficult to follow. To solve that specific point we propose to switch the following sentence : “We confirmed the absence of an additional divergent amh gene in *D. pectoralis* by searching sex-specific Pool-seq reads from 30 males and 30 females. In addition, only one amh homolog was found in an ongoing genome assembly project with long-reads for a male *D. pectoralis* (personal communication Y. Guiguen). “ just after the sentence stating that “In the more basally diverging species, *Dallia pectoralis* and *Umbra pygmaea*, we found only one *amh* gene in tissue-specific transcriptome databases (Pasquier et al., 2016).”

It is misleading to write "all Novumbra species" unless more than one was surveyed, which Table 1 suggests was not the case.

We agree that this would be misleading, but we have searched for this “all *Novumbra* species” mention in our current manuscript and we did not find it. The closest sentence is "… two well-supported clusters in all *Esox*, *Dallia* and *Novumbra* species" and even if our sentence is correct (as we do not refer to *Novumbra* species alone) we now changed it to "with two well-supported gene clusters among the *amh* sequences from Esocidae." in line with our answer to another comment below.

I also did not understand why the sentence says "However, two amh transcripts were readily identified in E. lucius (Pan et al., 2019), as well as in whole genome sequencing reads and assemblies (Supplementary file 1)". This sentence seems to confirm what was already stated above about E. Lucius, and perhaps it should be moved to an earlier place, saying "Moreover" rather than "However".

We agree that this sentence, that was changed between our initial submission and this first revision, is not understandable like it is written. We have now changed it to “Two *amh* genes were found in all surveyed *Esox* and *Novumbra* species. In more basally diverging species i.e., *Dallia pectoralis* and *Umbra pygmaea*, only one *amh* gene was found in both species in tissue-specific transcriptome databases (Pasquier et al., 2016), while two *amh* transcripts were readily identified in *E. lucius* (Pan et al., 2019)”

However, the next paragraph says that phylogenetic trees of the amh sequences showed "two well-supported clusters in all Esox, Dallia and Novumbra species", suggesting Dallia has both copies.

The reviewer is right about the interpretation of the phylogeny tree that the duplication of *amh* happened before the divergence of Esocidae and that *Dallia* had two copies. But from our extensive search, we only found one copy (in transcriptomes and in genomic reads of multiple individuals), and this result led to the idea that *Dallia* has lost its *amhby*. But we agree that this sentence might be confusing and we have hence changed it to “ These phylogenies provided a clear and consistent topology with two well-supported gene clusters among the *amh* sequences from Esocidae. “.

Some conclusions about these species are based on limited information in transcriptome databases, and some on other information, and the evidence should be more clearly and coherently explained.

As the reviewer suggested above we have moved these additional evidences just after the transcriptome evidence. We hope it will be easier to follow this way.

At this point, if I have guessed the correct meaning, I think that all Esox species, plus one species each of the genera Novumbra (but not Dallia and Umbra), have sequences that cluster with the E. lucius amha one, and are likely to be orthologs of this gene, and also sequences that cluster with the E. lucius amhby one, suggesting that the amh gene duplication preceded the split between the genera Esox and Novumbra, but perhaps occurred after the split from U. pygmaea (and perhaps Dallia). The text says that the "duplication happened after the divergence of Esocidae and Umbridae lineages", which suggests that Novumbra (with both copies) is not in the Umbridae.

With all respect we have to say that the reviewer has wrongly guessed the meaning. *Novumbra* is not an Umbridae and our sentence is totally right. We think that this comment is mainly due to the fact that reviewer #3 had problems following the track of the Esociformes phylogeny despite our efforts to be clear on that (Figure 1 and its corresponding legend). As we understand that it may be difficult (*Novumbra* sounds like *Umbra* and might suggest that it is also a member of Umbridae) and that this difficulty may be also shared by many other readers we modified this sentence to "duplication happened after the divergence of Esocidae (including *Esox*, *Dallia* and *Novumbra*) and Umbridae (including *Umbra*) lineages”

Please can the relationships be made clear before the results from them are described.

We have always been concerned about this potential difficulty and this is why we provided a main figure in our first submitted version, to better explain these phylogenetic relationships. No comments have been made by any reviewers, including reviewer #3, on this point during the first revision step. We believe phylogenetic relationships are clearly presented in the text. The main figure of the manuscript (Figure 1) should remove any doubts if any.

Clearly, the important questions are now whether the amhby copies are sex-determining genes in all species that carry the duplicate copy, and whether they acquired this function in the common ancestor of Esox and Novumbra, and retained this ever since. To test this, the study tested whether amhby was found in all males and absent from all females, in species additional to the E. lucius sample previously studied.The tests indeed detected associations between the male phenotype and the presence of amhby in most species of Esox (Table 1), but the associations were absent or weak in some Esox samples, including some North American E. lucius populations, and incomplete in the Novumbra species sample.

We agree and this is exactly what we did and wrote. We are very aware of the weak association due to low sample size for two of the *Esox* species for which our samples were obtained from museum collection. The reason for this low sample size was explained in the main text: “For two recently described species, *E. cisalpinus* and *E. aquitanicus* (Denys et al., 2014), we had insufficient samples with clear species and sex identification for a decisive result”. As the reviewer mentioned, in *Esox lucius* we actually showed here that *amhby* is lost in all North American species investigated outside of Alaska. So there is no association between amhby and sex phenotype. The entire section “Some populations of *Esox lucius* lost their Y chromosome and ancestral master sex determining gene” in the Results section is dedicated to this case study on these populations.

Overall, it is clear that the duplication is ancient, and pre-dates the split between Esox and Novumbra, but there does not seem to be strong evidence for the conclusion that it has maintained a male-determining function since the duplication occurred. The alternative cannot be excluded that the duplicate copy has acquired a sex-determining function independently in Esox and Novumbra, as well as losing amhb several times. In fact, this study seems a nice example of how much evidence is needed before one can be sure that a sex-determining gene has been maintained for a long time.

This is correct that we cannot completely exclude that *amhby* has acquired its sex determining function independently in *Esox* and *Novumbra*, but this alternative hypothesis is not the more parcimonious one. However as we agree that it could be still a possibility we introduced a short sentence in the Discussion stating that this is an alternative hypothesis clearly mentioning that this hypothesis is less parsimonious than the first one “Although we cannot rule out the possibility that *amhby* was recruited independently as the MSD gene in *Esox* and in *Novumbra,* our results suggest the more parsimonious hypothesis that *amhby* likely acquired an MSD function before the diversification of Esocidae at least 56 Mya.”.

The claim that the same region has remained as an undifferentiated sex-determining or Y-linked region for a long evolutionary time comes from the Pool-Seq analysis of E. masquinongy, one of the species with an incomplete association between the presence of a copy of the amh duplicate, to compare its size and location (as a putative sex locus) with that of E. lucius. Male-specific variants (described rather vaguely as "sex-specific heterozygosity") were detected in just a single genome region of less than 50 kb, and the region is homologous to the proximal end of the E. lucius LG24, where its sex-determining locus is located. This indeed suggests that both E. lucius and E. masquinongy have a physically small male-determining locus at the same location. However, it appears to have a weakened or partially lost function in E. masquinongy.

Yes, we agree. The *amhby* gene seems to have lost its association with male phenotype in some populations of *E. masquinongy*, like in some populations of *E. lucius*. Both species show population level variation in their SD systems. However, for comparison with the *E. lucius* sex locus, we selected a *E. masquinongy* population where amhby is associated with male phenotype. This association was actually also confirmed by the specific mapping of males-only reads on this *amhby* genomic region. Furthermore, although we are comparing one population from *E. masquinongy* and one population from *E. lucius*, the divergence time is comparable to what has been estimated between the two species and supports our result that homologous regions are constituting the SD locus in the two species with 50 million year of divergence.

However, the results do not tell us that they the male-determining function is ancestral, rather than independently evolved, in Esox and Novumbra. Indeed, subsection “Evolution of the structure of the amhby gene in the Esocidae” reports information that may suggest independent evolution in E. niger (in which amhby is strongly associated with maleness) and N. hubbsi. In both, the predicted Amhby protein is truncated in regions known to be important for this gene's function, but in E. niger the truncation is in its C-terminal part, whereas it affects the N-terminal part of the N. hubbsi sequence, and exon 1 encodes only eight amino acids, with no homology to the amino acid-sequence in other Esocidae.

As we explained above, we didn’t reject this alternative hypothesis based on our data, but considered the “ancestral MSD gene” as the more parsimonious scenario. As mentioned above we now provide a short sentence in the Discussion stating that this is an alternative hypothesis clearly mentioning that this hypothesis is less parsimonious than the first one.

For the other comments on truncated Amhby in E. niger and N. hubbsi we agree with reviewer #3. All what the reviewer mentioned is actually written in our manuscript and we do not see any question in this comment.

It is not explained whether the E. lucius amhby copy is complete, or how it differs from amha.

The characterization of *amhby*, its comparison with *amha*, and the functional demonstration of its role in the initiation of male sex determination in *E. lucius* is based on our already published results (Pan et al., 2019).

On the interpretation that these changes evolved independently, the similarity of other parts of the sequence (which provides the signal in the phylogeny) is potentially rather misleading.

This similarity is undoubtedly the result of common origin (as seen in the phylogeny as the topology of the tree is also in agreement with the taxonomy tree) and thus confirmatory rather than misleading.

The manuscript also reports evidence that sex-determining systems evolved independently in the 2 species that do not have the amh duplicate, D. pectoralis and U. pygmaea.It then turns to plausible suggestions about why Esox species might lose their male-determining factor. This section is rather unorganised, and should be greatly shortened to make clear what hypotheses were tested, and what the results were.

In our revision, we have already substantially restructured the Discussion following the suggestions from all three reviewers. As only reviewer #3 asked for additional re organizations (that will somehow also disorganize the current version and disrespect the agreement of the two other reviewers with this version) we would prefer keeping our discussion as it is.

Loss of genetic sex-determination is plausible in colonizing or low density situations, and the findings should be related to this concept.

This concept of a strong bottleneck effect in small founding populations during post-glaciation re-colonization is the main hypothesis we presented for the loss of *amhby* in some North American populations of *E. lucius* (see our Discussion section). A similar hypothesis could be eventually made for the *Dallia* transition but we feel that we don't have enough data to support this hypothesis in this species. The case of *Umbra* should not be considered as a transition, more like an ancestral state, followed by the emergence of a new SD system and eventually some transitions (in Esocidae). We agree that “the loss of genetic sex-determination is plausible in colonizing or low density situations” but because of that lack of data support in *Dallia* and also because our Discussion was already considered too lengthy, we do not want to add additional text on our Discussion on this point.